# Insights into azalomycin F assembly-line contribute to evolution-guided polyketide synthase engineering and identification of intermodular recognition

Guifa Zhai [1,2,5], Yan Zhu[1,2,5], Guo Sun [1,2], Fan Zhou[1,2], Yangning Sun[1,2], Zhou Hong[1,2], Chuan Dong[1,2], Peter F. Leadlay [3], Kui Hong[1,2], Zixin Deng [1,2], Fuling Zhou[1] & Yuhui Sun [1,2,4] ✉

Modular polyketide synthase (PKS) is an ingenious core machine that catalyzes abundant polyketides in nature. Exploring interactions among modules in PKS is very important for understanding the overall biosynthetic process and for engineering PKS assembly-lines. Here, we show that intermodular recognition between the enoylreductase domain $ER_{1/2}$ inside module 1/2 and the ketosynthase domain $KS_3$ inside module 3 is required for the cross-module enoylreduction in azalomycin F (AZL) biosynthesis. We also show that $KS_4$ of module 4 acts as a gatekeeper facilitating cross-module enoylreduction. Additionally, evidence is provided that module 3 and module 6 in the AZL PKS are evolutionarily homologous, which makes evolution-oriented PKS engineering possible. These results reveal intermodular recognition, furthering understanding of the mechanism of the PKS assembly-line, thus providing different insights into PKS engineering. This also reveals that gene duplication/conversion and subsequent combinations may be a neofunctionalization process in modular PKS assembly-lines, hence providing a different case for supporting the investigation of modular PKS evolution.

Modular polyketide synthases (PKSs) are highly sophisticated systems responsible for the biosynthesis of numerous natural products with diverse structures, and biological activities, which have been widely used as invaluable pharmaceuticals, agrochemicals, and veterinary agents[1]. The set of domain units operating in modular PKSs in control of one round of chain extension is termed a module, in which acyltransferase (AT), acyl carrier protein (ACP), and ketosynthase (KS) domains are essential for chain-elongation, while processing domains, such as ketoreductase (KR), dehydratase (DH), and enoylreductase

(ER) domains, may also reside in the module to adjust the extent of reductive modification of the β-keto group of the chain-elongation intermediate appropriately[2,3]. In addition, the thioesterase (TE) domain fused at the end of a PKS assembly-line hydrolyzes or macrocyclizes the fully elongated polyketide chain, promoting its release[4,5].

These different domain organizations generally allow for four types of modules and various permutations of domains with different substrate and stereo catalytic specificities, such as the malonyl-

[1]Department of Hematology, Zhongnan Hospital of Wuhan University, School of Pharmaceutical Sciences, Wuhan University, 430071 Wuhan, People's Republic of China. [2]Key Laboratory of Combinatorial Biosynthesis and Drug Discovery (Ministry of Education), Wuhan University, 430071 Wuhan, People's Republic of China. [3]Department of Biochemistry, University of Cambridge, Cambridge CB2 1GA, United Kingdom. [4]Wuhan Research Center for Infectious Diseases and Cancer, Chinese Academy of Medical Sciences, 430071 Wuhan, People's Republic of China. [5]These authors contributed equally: Guifa Zhai, Yan Zhu. ✉e-mail: yhsun@whu.edu.cn

CoA-specific ATa domain, methylmalonyl-CoA-specific ATp domain, and S/R-configuration-specific KR and ER domains, so that a six-module PKS assembly-line systems with differing the combination of these variants would theoretically be able to yield more than 100,000 polyketides[6,7]. The molecular recognitions between modules play vital roles in the organization of the modular PKS assembly-line, and ensure translocation of the nascent polyketide chain in a precise manner[1]. The mechanism by which modules recognize each other in a PKS assembly-line has been gradually revealed. (i) The interaction of the docking domains on the C-terminus of the upstream module and the N-terminus of the adjacent downstream module guarantees the specific recognition between the corresponding modules[8–10]. (ii) The intermodular recognition of an ACP and the downstream catalytic KS domain controls the efficient translocation of the growing polyketide chain for chain-elongation by the next module[11–13]. In most cases, these intermodular interactions control the linear arrangement and intermediate transfers of the modular PKS, thereby preventing promiscuous modification of intermediate chain and maintaining fidelity and efficiency in the chain extension process. Moreover, intermodular interactions also explain why the loss of function domain in a module tends to be accompanied by a downstream domain stuttering or action of a free-standing enzyme rather than domains in other modules in the assembly-line[14–16]. However, in our previous work[17–19], we uncovered a cross-module enoylreduction in which a switchable ER domain in module 1/2 acts in trans to catalyze the enoylreduction within the

module 3 in azalomycin F (AZL) PKS assembly-line (Fig. 1). Similar trans-acting catalytic effects have been found in the biosynthesis of aureothin[20,21], but the mechanism remains unclear. Unlike the aureothin pathway[21] in which the AT domain iteratively loads the extender unit onto successive modules in an intact PKS protein, module 1/2 and module 3 in AZL biosynthetic pathway are encoded respectively in two separate PKS genes (Fig. 1). Notably, we previously provided in vitro evidence showing that the $ER_{1/2}$-$KR_{1/2}$ didomain comprises the minimal cassette required for cross-module enoylreduction[19]. Therefore, we propose that the $ER_{1/2}$ domain in modules 1/2 and 3 are arranged into a hybrid for full enoylreduction via cross-module interactions.

In this work, considering that the recognition between modules plays a pivotal role in PKS catalysis, we demonstrate that the specificity of $ER_{1/2}$ is essential for cross-module enoylreduction. Moreover, we apply evolution-guided engineering of AZL PKS assembly-line and show that the divergent evolution of homologous modules 3 and 6 might be the reason for the functional loss of cross-module enoylreduction, leading us to explore the function of each domain in modules 3 and 6 through domain replacement to track the evolutionary process and decipher the mechanism of cross-module enoylreduction. We find that only in the presence of $KS_3$ can the $ER_{1/2}$ domain catalyze the cross-module enoylreduction. This work dramatically expands our understanding of PKS assembly-lines and provides different strategies for PKS assembly-line engineering. Moreover, it also provides direct experimental evidence for evolutionary development of PKS assembly-lines.

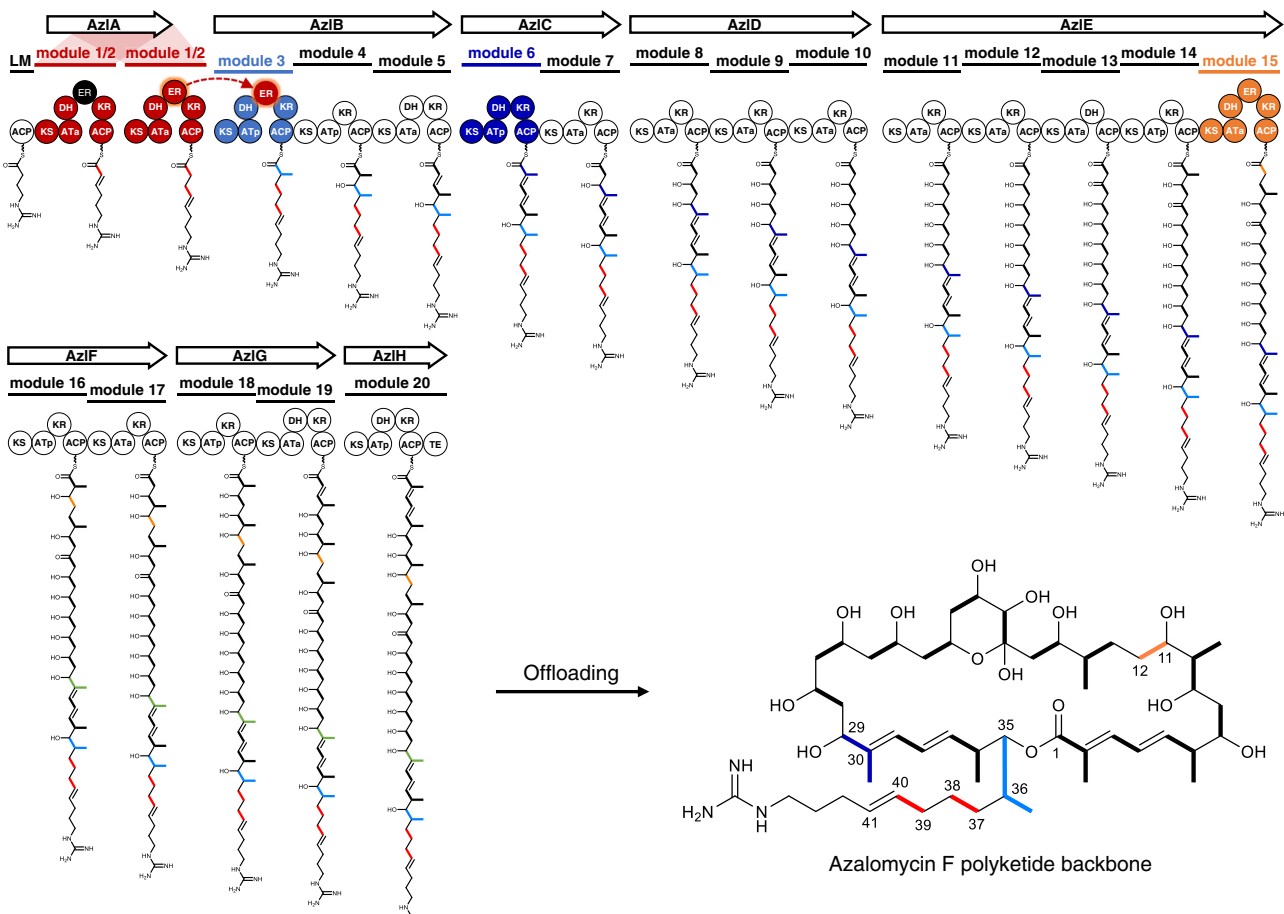

**Fig. 1 | Diagram for AZL polyketide backbone biosynthesis.** Each circle represents an enzymatic domain in the PKS, in which particularly ATa indicates the domain loading a malonyl-CoA to build an acetate unit (single bold line) and ATp to load a methylmalonyl-CoA to build a propionate unit (bold fold line), respectively. The modules and domains involved in this study are highlighted in colors. The red circle labeled with ER in module 3 indicates the missing enoylreductase domain whose enoylreduction is supplied in trans by the ER from module 1/2 and represented by dotted arc arrows. The black circle means that the $ER_{1/2}$ is non-function during the first round elongation of two successive iterations.

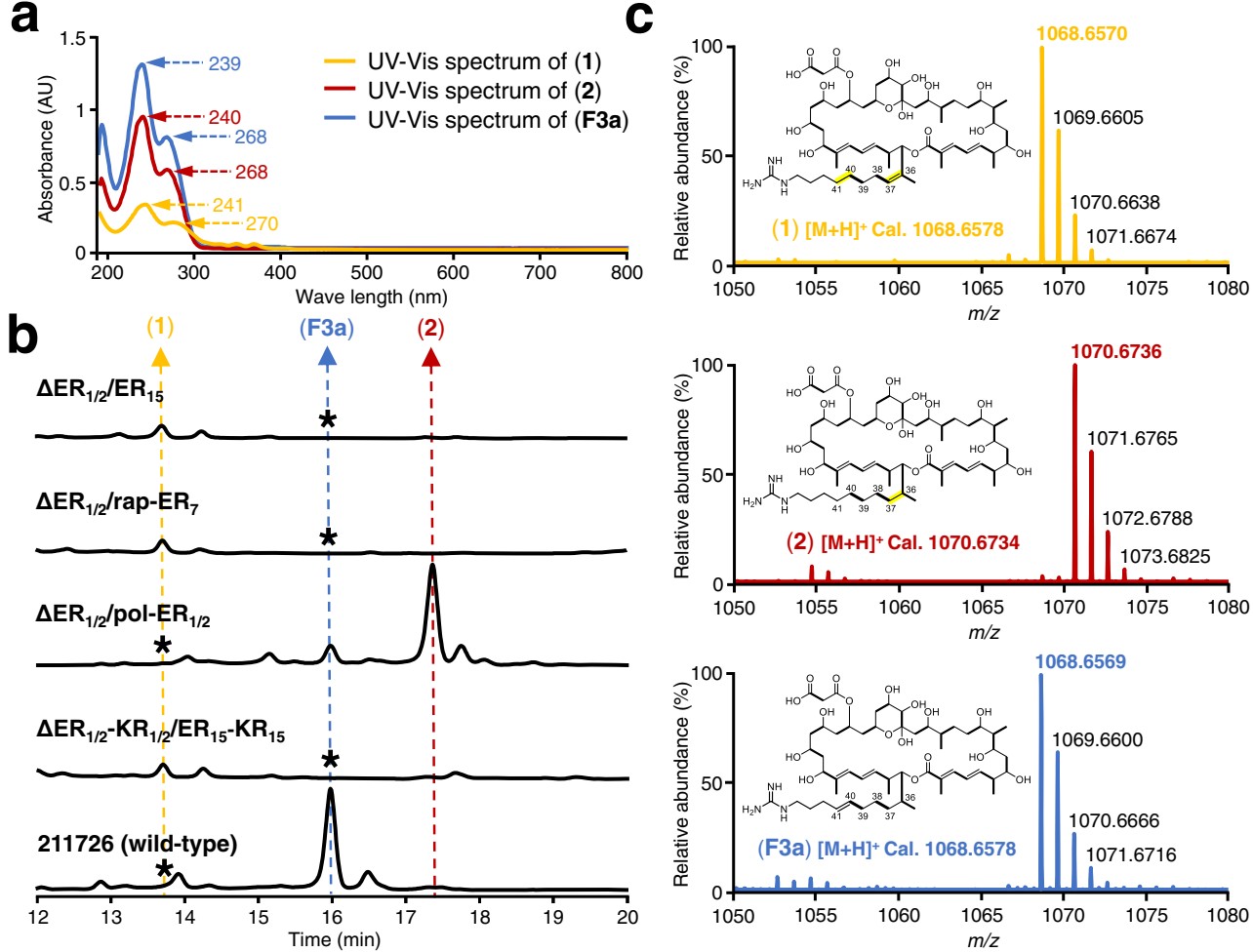

**Fig. 2 | Product analysis of strains with ER domain replacements. a** UV/Vis spectra of **F3a**, **1** and **2**. AU is absorbance unit. **b** HPLC analysis at λ = 254 nm. The same compounds are indicated by the dotted line with same color and no corresponding components detected are indicated by the asterisks. **c** LC-ESI-HRMS analysis of **F3a**, **1** and **2**. The spectra were extracted at $m/z$ ([M + H]$^+$) values equal to 1068.6578, 1068.6578, and 1070.6734, corresponding to compounds **F3a**, **1**, and **2**, respectively. Each experiment was repeated at least three times independently with similar results.

## Results

### The function-specific interaction between ER$_{1/2}$ and module 3 is crucial for non-canonical cross-module enoylreduction

The remarkable catalytic functions of modular PKSs are usually controlled by the interactions between domains and modules. To explore the correlation between the unusual behaviors and characteristics of ER$_{1/2}$, we replaced ER$_{1/2}$ in vivo with ER$_{15}$, the other canonical enoylreduction domain presented in module 15 of the AZL PKS (Fig. 1 and Supplementary Fig. 1). High-performance liquid chromatography (HPLC) and liquid chromatography coupled with electrospray ionization high-resolution mass spectrometry (LC-ESI-HRMS) analysis of the fermentation products showed that the replacement mutant ΔER$_{1/2}$/ER$_{15}$ completely abolished the production of AZL. A significant new peak (compound **1**) with molecular weight and ultraviolet/visible (UV/Vis) absorption spectrum similar to AZL **F3a** was identified (Fig. 2), suggesting that it could be an AZL **F3a** analog. Meanwhile, a trace of an unexpected peak (compound **2**) was also found. It had a similar UV/Vis absorption spectrum with AZL but 2 Da greater than AZL **F3a** in molecular weight (Fig. 2a, c). To characterize the structure of compounds **1** and **2**, we performed large-scale fermentation. Their structures were determined through a comprehensive analysis of their 1D and 2D NMR spectra (Supplementary Tables 1 and 2 and Supplementary Notes 1 and 2). For compound **1**, a new double bond was evident at C36-C37, and the bond at C40-41 was reduced to a single bond, in

contrast to that in AZL **F3a**, as confirmed by the key HMBC correlations of H-35/C-1, C-33, C-34, C-50, C-36, C-37, and C-51 (Supplementary Table 1 and Supplementary Note 1). In compound **2**, the double bond at C36-C37 was reduced to a single bond, similar to that in AZL **F3a**, as verified by the correlations between H-35/C-1, C-33, C-34, C-50, C-36, C-37, and C-51 (Supplementary Table 2 and Supplementary Note 2). The above results showed that, in contrast to the ER$_{1/2}$ domain, ER$_{15}$ catalyzed the successive enoylreduction during the first two rounds of chain elongation, which indicated that the non-canonical enoylreduction owed to the peculiarity of ER$_{1/2}$ itself. Comparing the structure of compounds **2** and **1**, we speculated that reduction at C36-C37 in **2** could result from non-specific substrate processing by ER$_{15}$ during the third round of elongation or through intrinsic cross-module enoylreduction by ER$_{15}$.

To test whether the production of compound **2** was due to the particular properties of ER$_{15}$, we replaced the ER$_{1/2}$ in AZL wild-type with the exogenous and canonical ER domain of module 7 from the rapamycin PKS assembly-line and a highly similar ER$_{1/2}$ domain from polaramycin PKS assembly-line, respectively (Supplementary Figs. 1 and 2b). The structure of the macrocyclic antifungal polaramycin and the module organization of its PKS assembly-line closely resembles that of AZL, and the pol-ER$_{1/2}$ domains are naturally used for the enoylreductions required during the first three rounds of chain extensions (Supplementary Fig. 2b). Therefore, the ER domain

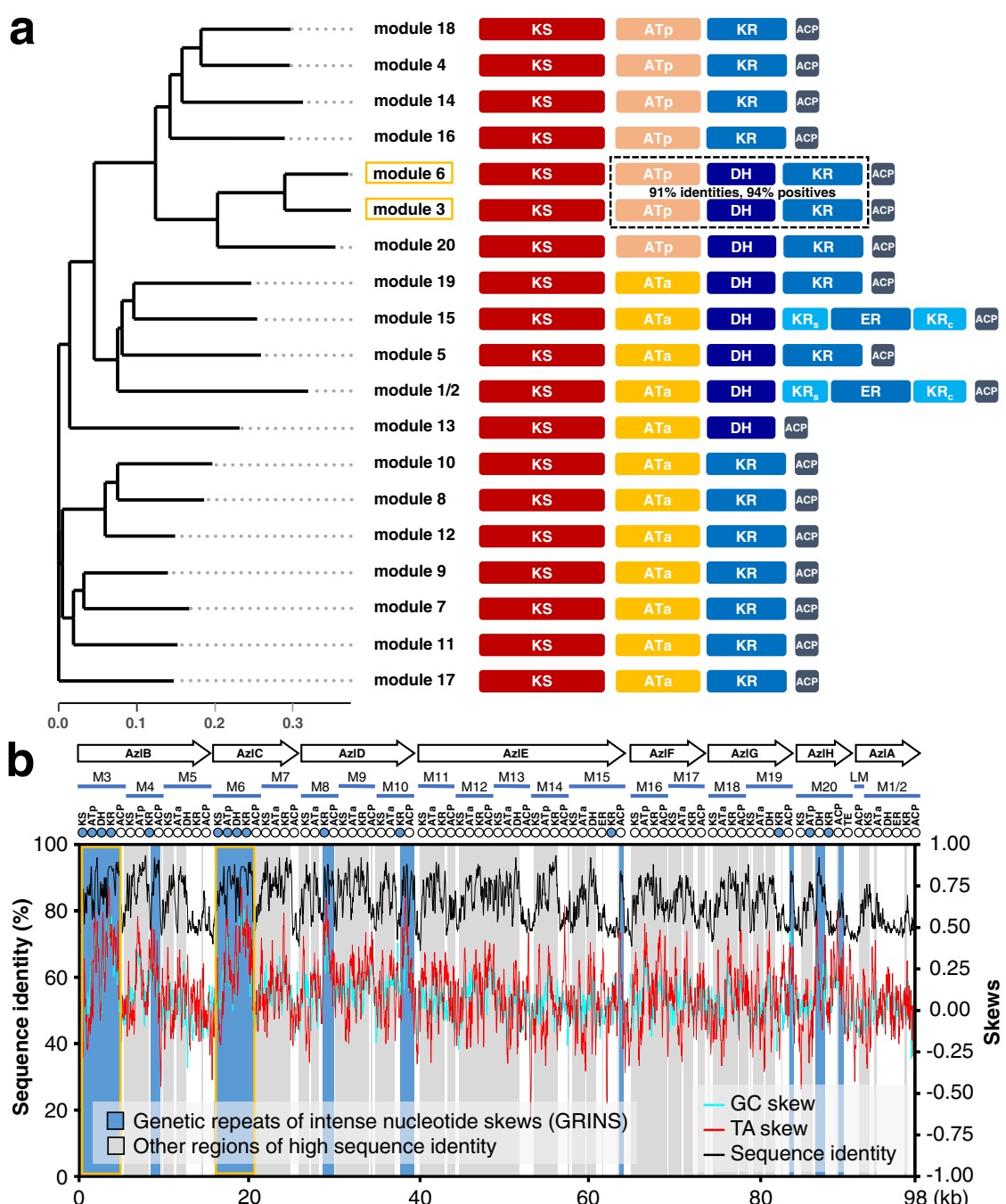

**Fig. 3 | Phylogenetic and genetic repeats of intense nucleotide skew (GRINS) analysis of AZL PKS. a** Phylogenetic analysis of AZL PKS modules. Each bar represents a domain in PKS, with ATa the domain that loads a malonyl-CoA to build an acetate unit and ATp the domain that loads a methylmalonyl-CoA to build a propionate unit. The numbers in the dotted frame show identities and positives of the ATp-DH-ER tridomain between modules 3 and 6. **b** GRINS in AZL PKS.

Duplicated regions (>80% DNA sequence identity to another region within the same PKS) containing the intense GC and AT skews are annotated as GRINS. Although sequences in AZL PKS assembly-line are highly similar (the black curve), GRINS only are present in several regions (the blue shade). The highly identical GRINS in module 3 and module 6 are outlined in yellow.

replacement mutant $\Delta ER_{1/2}$/pol-$ER_{1/2}$ was ideally used here as a positive control. Remarkably, a tiny amount of compound **2** at a level similar to that by $\Delta ER_{1/2}$/$ER_{15}$ was produced by $\Delta ER_{1/2}$/rap-$ER_7$ (Fig. 2b), implying that $ER_{15}$ exhibited no specificity and any replacement of $ER_{1/2}$ in AzlA with any canonical ER domain, such as $ER_{15}$ and rap-$ER_7$, seems to be able to a slight extent at least to be able to catalyze the non-specific cross-module enoylreduction via an as yet unknown mechanism. A large yield of compound **2** was the main product of $\Delta ER_{1/2}$/pol-$ER_{1/2}$, which indicated that the particularity characteristics of $ER_{1/2}$ were

essential for this unusual catalysis and suggested the importance of interaction between $ER_{1/2}$ and its neighboring module 3 in cross-module enoylreduction. While a much lower AZL **F3a** also accumulated in the mutant $\Delta ER_{1/2}$/pol-$ER_{1/2}$, suggesting that pol-$ER_{1/2}$ was insufficient to catalyze the enoylreduction on all the nascent chain in the first round of chain elongation before translocation, which resulted in the accumulation of C40-C41 unsaturated product. To confirm the catalytic properties of pol-$ER_{1/2}$, we fermented the polaramycin-producing strain *Streptomyces hygroscopicus* LP-93[22]. Analysis of the

**Fig. 4 | Diagram showing azlB deletion when engineering the AZL PKS assembly-line and its product.** The dotted circle labeled with ER in module 3 represents the missing enoylreductase domain whose enoylreduction is supplied in trans by the ER from module 1/2. The asterisks indicate that no corresponding components were found. Each experiment was repeated at least three times independently with similar results.

fermentation products (Supplementary Fig. 3) resulted in the discovery of a small amount of product (compound **6**), in which pol-$ER_{1/2}$ was not working during the first round of chain extension. This result demonstrated that the catalysis by pol-$ER_{1/2}$ led to AZL **F3a** production by the mutant $\Delta ER_{1/2}$/pol-$ER_{1/2}$ (Fig. 2b). Although both $ER_{1/2}$ and pol-$ER_{1/2}$ catalyzed full enoylreduction on the nascent polyketide chain during second and third round extension, the substrate preferences during the first round of chain extension were mostly different, which might have resulted from divergent evolution between the parent and off-spring genes.

## Bioinformatics analysis indicated that modules 3 and 6 are evolutionarily homologous in the AZL PKS assembly-line

Typically, there is a binding site in a module to recruit the free-standing proteins needed to catalyze the modification of thiotemplate intermediates in modular assembly-lines. For example, the AT docking domain in trans-AT PKS assembly-lines serve as the binding site through which a discrete AT loads the extender unit to the module[23]. The X-domain in modular non-ribosomal peptide synthetase (NRPS) assembly-lines recruits cytochrome P450 oxygenases to catalyze the crosslinking of aromatic side chains during the biosynthesis of vancomycin and teicoplanin[24]. An ACP domain with an unusual tryptophan motif is needed to trigger the operation of the β-branching cassette in both trans-AT and cis-PKSs[25]. Moreover, the functional domains in modules can also function as a landing site. Thus, Wang and coworkers proved that a discrete ER LovC binds with the AT domain in LovB during lovastatin biosynthesis, selectively catalyzing the enoylreduction required at different stages of chain growth[26]. In the case of $ER_{1/2}$ cross-module enoylreduction involving module 3, a careful analysis of module 3 did not indicate that any particular domain was responsible

for the binding of $ER_{1/2}$ in module 3. We then compared the alignment of module 3 with modules 5, 6, 19, and 20 that all have the same domain organization (KS-AT-DH-KR-ACP) as module 3. Interestingly, we found that both modules 3 and 6 contained almost identical truncated KR domains, compared with other modules (ca. 50 amino acids) (Supplementary Fig. 4). The comparison also showed that the $AT_3$-$DH_3$-$KR_3$ tridomain exhibited an unusually high sequence identity with $AT_6$-$DH_6$-$KR_6$ (92% identity at the nucleotide level and 91% amino acid identity at the protein level) compared with the other modules, suggesting that these tridomains are evolutionarily homologous and that modules 3 and 6 may have coevolved.

To provide more evidence of modules 3 and 6 coevolution, a phylogenetic analysis of all domains and modules in AZL PKS was performed. Specifically, all the module sequences of AZL PKS were aligned using the muscle codon-based method; thereafter, a neighbor-joining tree (Jukes-Cantor distance model) of all the modules was constructed and visualized by MEGA-CC (version 11.0.11)[27] and ggtree[28]. As anticipated, the results showed that modules 3 and 6 were located on two branches of a single clade (Fig. 3a), along with pairs $KS_3$ and $KS_6$, $AT_3$ and $AT_6$, $DH_3$ and $DH_6$, $KR_3$ and $KR_6$, and $ACP_3$ and $ACP_6$ derived from modules 3 and 6 grouped together (Supplementary Fig. 5), which indicated that modules 3 and 6 show the closest evolutionary relationship among modules. This comparison of sequence similarity and an evolutionary analysis strongly supported the idea that tandem tridomains $AT_3$-$DH_3$-$KR_3$ and $AT_6$-$DH_6$-$KR_6$ originating from the same ancestral sequence (Supplementary Figs. 4 and 5). The phylogenetic tree also revealed another interesting finding: the topological structure of the tree was clustered into two parts (Supplementary Fig. 5), which was consistent with two categories of AT domains that can load two types of substrates in AZL PKS (the malonyl-CoA-specific

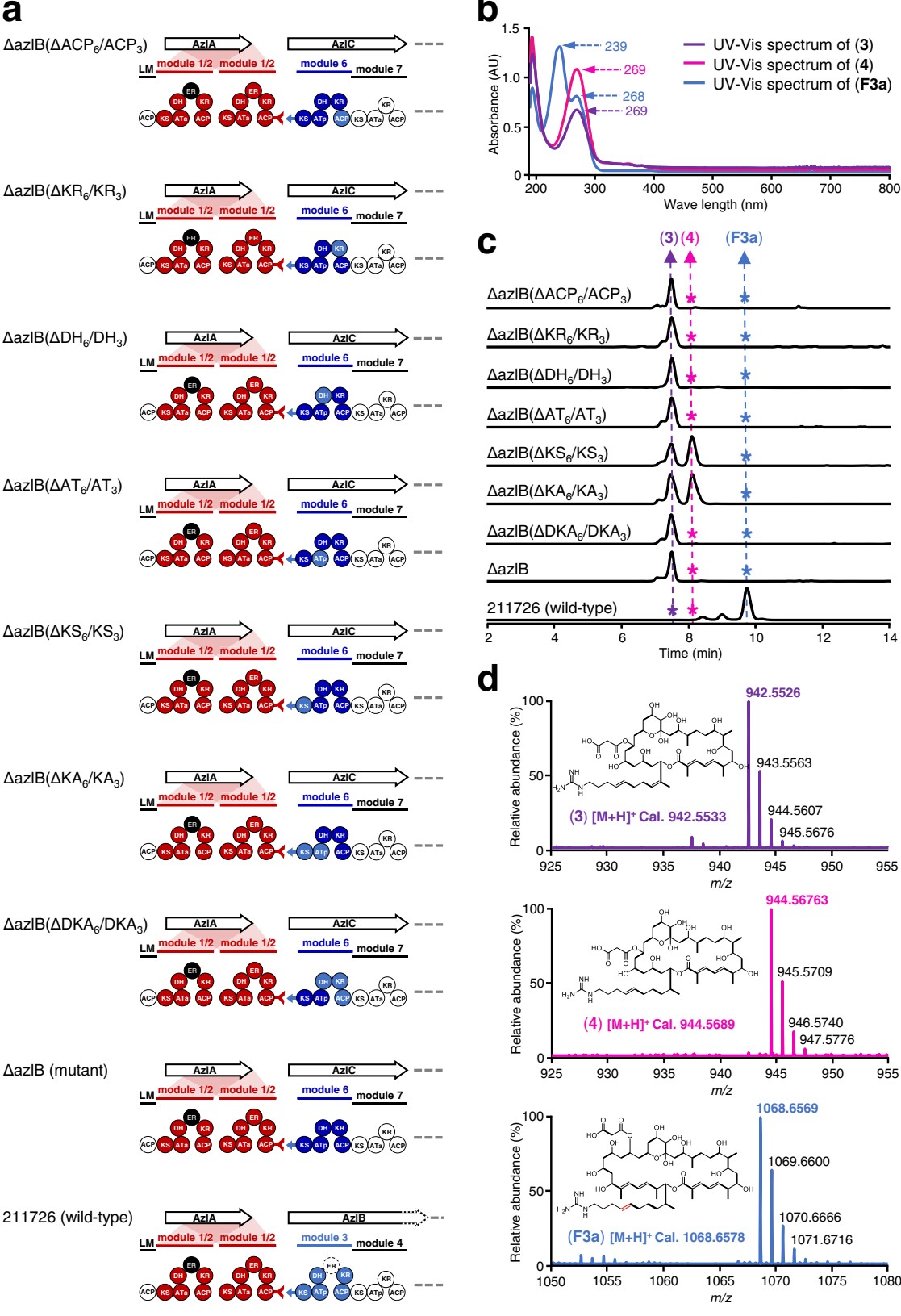

**Fig. 5 | Products analysis of domain replacement strains based on ΔazlB mutants. a** Diagram for domain replacements in ΔazlB mutants. The dotted circle labeled with ER in module 3 of AzlB represents the missing enoylreductase domain whose enoylreduction is supplied in trans by the ER from module 1/2. **b** UV/Vis spectra of **F3a**, **3** and **4**. AU is absorbance unit. **c**, **d** LC-ESI-H RMS analysis of **F3a**, **3** and **4**. The traces and spectra were extracted at *m/z* ([M + H]⁺) 1068.6578, 942.5533

and 944.5689 corresponding to compound **F3a**, **3** and **4**, respectively. The asterisks indicated that no corresponding components were found. For simplicity, KS-AT didomain and DH-KR-ACP tridomain were abbreviated as KA and DKA in **a** and **c**. Each experiment was repeated at least three times independently with similar results.

ATa domain and methylmalonyl-CoA-specific ATp domain). The relatively rapid evolution of ATp cluster compared to ATa cluster might explain their substrate biases. Notably, the sequence alignment showed that AT domains shared almost 90% sequence identity with each other in the same group, except for $AT_{1/2}$ and $AT_{10}$ (Supplementary Fig. 5), which suggested that the same group of AT domains have the same ancestor.

Recently, gene duplication and conversion have been proposed to explain the evolution of the PKS assembly-lines[29,30]. Nivina and coworkers showed that the gene conversion of genetic repeats of intense nucleotide skews (GRINS) recode and accelerate the diversification of the PKS assembly-line, and recombination between GRINS was also observed via a bioinformatics analysis[30,31]. The properties of higher nucleotide composition bias and lower sequence complexity of GRINS would easily make them to undergo gene conversion, and thus to recode a neighboring homolog at a very high frequency. Evolution insight into AZL PKS assembly-line showed that module 3 and module 6 are evolutionarily homologous. To further explore the possible evolutionary process of these modules, we performed the GRINS analysis of AZL PKS assembly-line. Based on the GRINS analysis of AZL PKS, we found two nearly identical GRINS in the KS-AT-DH-KR tetradomain in modules 3 and 6 (Fig. 3b), which further supported an evolutionary homology between modules 3 and 6. Accordingly, we postulated that the higher similarity of the AT-KR-DH tridomain between modules 3 and 6 results from gene duplication or conversion, suggesting a clear possibility for neofunctionalization. Moreover, the higher similarity between these modules indicates the possibility of engineering the AZL PKS assembly-line guided by evolution information.

## Evolution-guided engineering of the AZL PKS assembly-line

Recombination events are implicated in the evolution of modular PKS assembly-lines, especially in trans-AT PKS systems[32,33], but there is less evidence for similar mechanisms in cis-AT PKS evolution[7,31,33,34]. In the light of the results of bioinformatics analysis of the AZL PKS assembly-line and the accelerated evolution via module recombination within the rapamycin and tylosin PKS systems[31], we constructed a knockout mutant of *azlB* of the AZL PKS (Fig. 4) with the purpose of emulating the evolutionary recombination of homologous modules. This would not only enable us to test whether $ER_{1/2}$ can be recruited by module 6 to catalyze cross-module enoylreduction but also to further study the role played by gene conversion in PKS evolution. Considering the recognized specificity of its docking domain, we knocked out *azlB* while retaining its *N*-terminal docking-domain sequence (Fig. 4).

The β-hydroxyl group produced during the fourth round of chain elongation determines the macrocyclisation of AZL in the wild-type. As comparison showed that the domain organization in the first four modules of both wild-type and the ΔazlB mutant are essentially identical (Fig. 4), it seemed very likely that the resulting shortened polyketide chain might also be released via the hydroxy group presumably generated by the module 7 in the fourth round of chain elongation, creating the identical length guanidyl side chain which targets the lipoteichoic acid to induce autolysis of methicillin-resistant *Staphylococcus aureus*[35,36].

HPLC and LC-ESI-HRMS analyses revealed a new peak compound **3** in the ΔazlB mutant. However the molecular weight was 2 Da less than that predicted (Fig. 5d), suggesting that the α,β-double bond produced in the third round of chain elongation was preserved in the product. This indicated that, although modules 3 and 6 might be evolutionarily homologous, module 6 cannot function in the same way as module 3, which recruited $ER_{1/2}$ for enoylreduction on its nascent polyketide chain. To confirm this finding, we performed large-scale fermentation to characterize the structure of compound **3**. The structure was determined based on the molecular ion peak at *m/z* 942.5526 in the LC-ESI-HRMS spectrum (Fig. 5d) and comprehensive

analysis of 1D and 2D NMR spectra (Supplementary Table 3 and Supplementary Note 3). The double bonds at both C30-C31 and C34-C35 were confirmed by the crucial HMBC correlations of H-29/C-28, C-27, C-30, C-31, C-43 and H-33/C-31, C-32, C-34, C-35 (Supplementary Table 3 and Supplementary Note 3). Thus, we not only achieved the predicted evolution-oriented AZL PKS engineering but also provided evidence that the recognition of the docking domain between modules 1/2 and 3 cannot be the only thing involved in the process of cross-module enoylreduction. More significantly, this result demonstrated that some divergence between modules 3 and 6 must be the reason for $ER_{1/2}$ recruitment which is not present in module 6.

## The $KS_3$ domain is essential for $ER_{1/2}$ catalysis of cross-module enoylreduction

In order to characterize the difference between modules 3 and 6, further elucidate the mechanism of cross-module enoylreduction and ultimately track module evolution; specifically, we replaced each domain of module 6 with the corresponding domain of module 3 in the ΔazlB mutant (Fig. 5 and Supplementary Fig. 6). To investigate the possibility that cross-module enoylreduction could be due to interactions among multiple domains, based on the structural cognition of modular PKS, we constructed KS-AT didomain and DH-KR-ACP tridomain replacement mutants of ΔazlB (Fig. 5 and Supplementary Fig. 6). The HPLC and LC-ESI-HRMS results revealed that a new peak representing a compound with a molecular weight that was 2 Da more than compound **3** accumulated only in $KS_3$ and $KS_3$-$AT_3$ replacement mutants ΔazlB(Δ$KS_6$/$KS_3$) and ΔazlB(Δ$KS_6$-$AT_6$/$KS_3$-$AT_3$) (Fig. 5). The structure of compound **4** was determined from the molecular ion peak at *m/z* 944.5676 in the ESI-HRMS spectrum (Fig. 5d) and a comprehensive analysis of its 1D and 2D NMR spectra (Supplementary Table 4 and Supplementary Note 4). The bond at C30-C31 in compound **4** was reduced relative to that in compound **3**. This was verified by the important HMBC correlations of H-29/C-28, C-27, C-30, C-31, C-43 (Supplementary Note 4). These results revealed that chimeric module 6 recruited $ER_{1/2}$ to achieve cross-module enoylreduction only in the presence of $KS_3$. However, the continued presence of a small amount of compound **3** in the KS and KS-AT replacement mutant, suggests that the complete cross-module enoylreduction in wild-type requires the action of other factors, which still needs to be further elucidated. The hypothesis of gatekeeper domains could explain why certain amount of compound **3** still accumulated in KS replacement mutant, due to different substrate preference between the module 3 downstream gatekeepers $KS_4$ in wild-type and $KS_7$ in ΔazlB. Nevertheless, these results conclusively demonstrated that the intermodular interaction between $ER_{1/2}$ and $KS_3$ was essential for cross-module enoylreduction.

## Plausible mechanism of cross-module enoylreduction

The above $ER_{1/2}$ replacement results showed that the function of $ER_{1/2}$ cannot be complemented by $ER_{15}$, so the properties of $ER_{1/2}$ are essential for cross-module enoylreduction. However, the biosynthetic rationale of the trace amounts of compound **2** produced in Δ$ER_{1/2}$/$ER_{15}$ and Δ$ER_{1/2}$/rap-$ER_7$ remains unknown. The previous in vitro assay results demonstrated that the $ER_{1/2}$-$KR_{1/2}$ domain was the minimal catalytic cassette required for cross-module enoylreduction[19], which led us to assume that $KR_{1/2}$ may also interact with module 3, facilitating $ER_{15}$ in achieving non-specific cross-module enoylreduction in the mutants. To test this, we constructed the mutant Δ$ER_{1/2}$-$KR_{1/2}$/$ER_{15}$-$KR_{15}$ (Supplementary Fig. 7). However, traces of compound **2** remained, which suggested other factors would be the reason for the accumulation of a small amount of cross-module enoylreduction product (Fig. 2b).

$ACP_{1/2}$ may also interact with $KS_3$ to assist in the non-specific cross-module enoylreduction. However, Lowry and coworkers[37] demonstrated that when the nascent polyketide chain was tethered to the ACP domain, and the cognate KS domain was in a closed state to

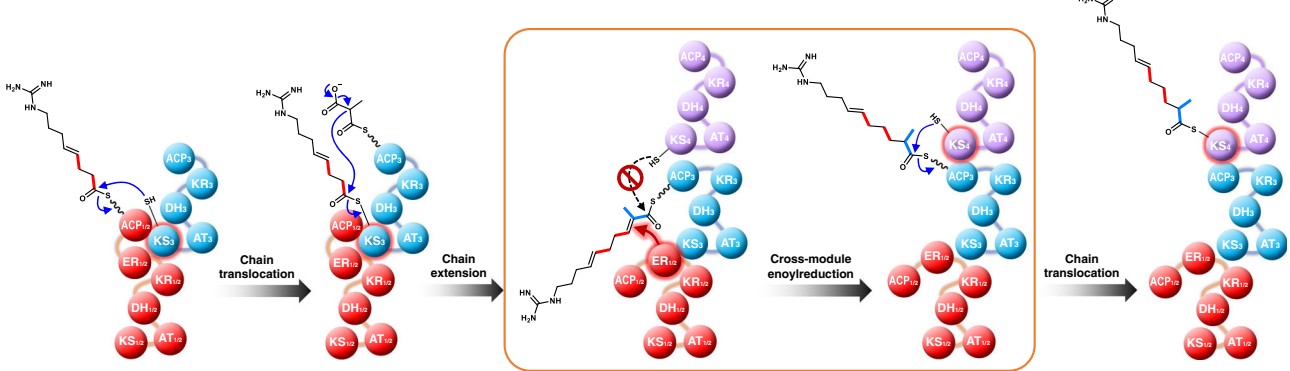

**Fig. 6 | Illustration of the proposed mechanism of cross-module enoylreduction in AZL biosynthesis.** The core of the proposed mechanism is highlighted in the orange box, indicating that only when completion of cross-module enoylreduction, can the reduced and preferred intermediate tethered onto ACP3 transfer to KS4 of module 4.

prevent acylation by the growing intermediate tethered to the upstream ACP domain until the nascent polyketide chain was transferred onto the downstream module[37]. This turnstile mechanism has been further confirmed via the characterization of the dynamic 3D structure of 6-deoxyerythronolide B synthase module 1[38]. Therefore, it was determined that the KS3 domain should be in a closed state and would not interact with the ACP1/2 domain during the cross-module enoylreduction of the intermediate tethered to the ACP3 domain, which eliminates the possibility that the interaction of the ACP1/2 and KS3 domains is involved in cross-module enoylreduction. Additionally, the absence of any compound **4** in ΔazlB indicated that the docking domain could not be only thing involved in the cross-module enoylreduction.

Analysis of the AZL PKS assembly-line revealed that KS4 preferred an α,β-saturated nascent chain 1 (Fig. 1). We speculated that the replacement of ER1/2 with ER15 would therefore cause the loss of cross-module enoylreduction ability of the engineered module 1/2 and would result in a critical delay of transfer of the unreduced α,β-unsaturated nascent chain tethered onto ACP3 under the strict substrate recognition of the gatekeeper KS4[39–42]. This would provide ER15 the opportunity to catalyze the non-specific cross-module enoylreduction to yield compound **2** albeit with low efficiency[39–42]. So, we infer that the gatekeeper KS4 may also contribute to cross-module enoylreduction, accounting for the accumulation of trace amounts of compound **2** in both the ΔER1/2/ER15 and ΔER1/2/rap-ER7 mutants (Fig. 2b). This hypothesis can also explain the reason that production level of compound **2** in ΔER1/2/pol-ER1/2 is similar to that of AZL **F3a** in wild-type, which was much higher than the level of compound **1** in ΔER1/2/ER15 (Fig. 2b). Taking the above results together, we proposed a plausible mechanism for cross-module enoylreduction. In this, KS3 is the anchor site for ER1/2 to catalyze the enoylreduction of the intermediate tethered to module 3 (Fig. 6), and KS4 acts as a gatekeeper inhibiting the transfer of the ACP4-bound intermediate until ER1/2-mediated reduction has taken place.

## Discussion

Modular PKS are arranged into the required assembly-line order via the docking domains to ensure efficient polyketide chain elongation and intermodular transfer. Thus, rewiring the PKS assembly-line by exchanging and recombining these recognition subunits is a promising strategy for generating novel polyketide products[43]. In the case of AZL PKS, we demonstrated an intermodular recognition to effect cross-module enoylreduction and achieved the gain-of-function of engineered module 6 in the recruitment of ER1/2, thus realizing cross-module enoylreduction. In addition, we provided evidence in favor of how the process of evolution PKS assembly-lines by gene duplication/

conversion offers the possibility for the reorganization of the PKS assembly-line and following gene change promotes diversity of function in PKS assembly-lines.

Great efforts have been devoted to achieving the rational engineering of modular PKS assembly-lines in the past three decades[44,45]. Although the mutual simulation between homologous PKS assembly-lines based on the comparative analyses has been proven to be a feasible engineering method[46–48], it is limited to PKS assembly-lines with homologous gene clusters. Meanwhile, this strategy only takes the homology of inter-PKS assembly-lines into account whereas there existed homologous regions within a PKS assembly-line. Therefore, an in-depth understanding of the catalytic mechanism and evolution process of polyketide synthase has theoretical guiding significance for engineering the PKS assembly-line in a universal strategy. Considering the broad distribution of homologous modules within a PKS assembly-line, especially in cis-AT PKS assembly-lines[30], we provided a proof-of-concept of an evolution-oriented engineering strategy for PKS assembly-lines via emulating the recombination process of homologous modules.

According to the results above, module 6 cannot function as does module 3 to recruit ER1/2 for cross-module enoylreduction. This outcome may be a result of the divergence of the KS domain, as indicated by the hypothesis that ATn-DHn-ERn-KRn-ACPn-KSn+1 can be a unit with a long evolutionary history[49,50]. In the ΔazlB(ΔKS6/KS3), the native interfaces between ACP1/2-KS3 and ACP6-KS7 were maintained, ensuring polyketide chain translocation efficiency. The enhancement of fermentation confirms that preserving the native interaction between ACPn and KSn+1 was important for efficient polyketide chain translocation.

In canonical modular PKS systems, domain and module engineering are usually performed to produce PKS assembly-lines to generate the desired products with broad-spectrum activities and diverse structures[51–56]. Problems such as genetic manipulations, cloning of large gene clusters, and screening double crossover mutants greatly hinder achieving engineering goals. In this work, we described an intermodular interaction that endows ER1/2 to carry out an in trans cross-module enoylreduction on its downstream module 3. This provides a completely different approach to PKS engineering. Notably, this work would lead us to quickly achieve the trans-acting modification of intermediates tethered to specific modules via discrete designable gene elements rather than the engineering of target modules in the future.

Several crystal structures of mammalian fatty acid synthase and fungal iterative PKS show that the ER domains are homodimers in these iterative multienzyme complexes[57,58]. In this work, the data showed that KS3 is the anchor for ER1/2, leading to the catalysis of cross-

module enoylreduction with 100% catalytic efficiency in the wild-type AZL PKS assembly-line, which led us to hypothesize that $ER_{1/2}$ seems to act as part of module 3 during the third round of chain elongation. In light of the structural information of CTB1 SAT-KS-MAT in which the starter unit acyltransferases (SAT) connected with the KS domain and the active site entrance of each SAT point into the reaction chamber of CTB1[59]. Similarly, we speculated that the conformation of $KR_{1/2}$ and $ER_{1/2}$ would also undergo a certain angle of rotation along the X-axis, resulting in the connection between $ER_{1/2}$ and $KS_3$ with the substrate tunnel orientation of $ER_{1/2}$ toward the reaction chamber of module 3 in the process of cross-module enoylreduction (Supplementary Fig. 8). Considering that KS domains are homodimeric in modular PKS[13,38,60–62], we thus postulated that the $ER_{1/2}$ domain might also be homodimeric when binding to $KS_3$ during the third round of chain elongation (Supplementary Fig. 8). The sequence alignment revealed another interesting point that there were two truncated KR domains in AZL PKS assembly-line (Supplementary Fig. 4). KR domain consists of two Rossman folds, which are often called catalytic subdomain and structural subdomain[63]. The catalytic subdomain contains the cofactor of NADPH for the intermediate keto reduction, and the structural subdomain does not bind NADPH. The modeling structure of $KR_3$ and $KR_6$ revealed that there was an obvious structural difference in truncated $KR_3$ and $KR_6$ in which two pairs of α-helix and β-fold are missing in the structural subdomain compared with other KR domains like $KR_5$ (Supplementary Fig. 9). However, the missed α-helix and β-fold only disrupted the Rossman fold and did not affect the folding of the structural subdomain, as they lie on the outmost side (Supplementary Fig. 9). Although the structural subdomain is very important in stabilizing the KR domain, it does not bind a NADPH making the Rossman fold seemly redundant. So, it is rational to evolve into a compact structure for KR domain without affecting the overall structure. Besides, we speculated that the shortened $KR_3$ would provide enough space for $ER_{1/2}$ to bind and catalyze such unusual cross-module enoylreduction. Although several modular PKS 3D structures (KS-AT-KR-ACP) have been elucidated via cryogenic electron microscopy[13,38,61,62], the great flexibility and large molecular weight have hindered elucidation of the structure of the fully reduced modules (KS-AT-DH-KR-ER-ACP). Nevertheless, our results provide the glimpse into the possible interaction mechanism of $ER_{1/2}$ and its partner $KS_3$.

In summary, this work elucidated a non-canonical pathway of the AZL PKS assembly-line and provided evidence for the intermodular recognition between $ER_{1/2}$ and module 3, while the $KS_4$ domain, by acting as a gatekeeper, indirectly enhances the particularity of ER in cross-module enoylreduction. Although structural evidence is needed to determine the precise molecular mechanism of this discovered cross-module enoylreduction, our findings expand understanding of modular PKS assembly-lines and provide insights into modular PKS engineering.

## Methods

### General methods
All chemical reagents and antibiotics were purchased from Sigma-Aldrich and Sangon Biotech. All molecular cloning reagents were purchased from New England BioLabs and QIAGEN. All oligonucleotide synthesis and DNA sequencing services were provided by Wuhan Tsingke. Bacterial strains, plasmids, and primers are summarized in Supplementary Tables 5–7, respectively.

### Fermentation conditions
*Streptomyces* sp. 211726 wild-type and mutant strains were developed in SFM solid medium (3.3% soya flour, 2% mannitol, 2.5% agar) for rejuvenation and conjugation with an additional 120 mM $CaCl_2$ at 28 °C for 5–7 days. A seed culture of *Streptomyces* was developed in TSBY liquid medium (3% tryptone soy broth, 0.5% yeast extract, 10.3% sucrose) at 28 °C and 220 rpm on a rotary incubator for 2–3 days to perform total DNA isolation and further fermentation. Fermentation of *Streptomyces* was conducted in SFM liquid medium (3.3% soya flour, 2% mannitol) at 28 °C and 220 rpm on a rotary incubator for 5–7 days.

### Construction of plasmids for $ER_{1/2}$ domain replacement
To construct the domain replacement mutant $\Delta ER_{1/2}/ER_{15}$ in vivo, three fragments including $ER_{15}$ sequence and two homologous arms induced by three pairs of primers $ER_{15}$-sw-F, $ER_{15}$-sw-R, $ER_{15}$-sw-L1, $ER_{15}$-sw-L2, $ER_{15}$-sw-R1, and $ER_{15}$-sw-R2 were firstly amplified from *Streptomyces* sp. 211726 genome DNA. Then, the *Streptomyces-E. coli* shuttle vector pYH7 digested by *Hin*dIII and *Nde*I were fused with the PCR products to conduct pWHU5033 by Gibson assembly. And similar procedures are performed to obtain pWHU5034, pWHU5035, pWHU5036 (Supplementary Table 6) with pairs of corresponding primers (Supplementary Table 7). Restriction endonuclease digestion and sequencing were applied to confirm the constructed plasmid.

### Phylogenetic tree construction
The sequences of AZL PKS were firstly aligned by using muscle codon-based algorithm. Hereafter, the distances for all sequence pairs were calculated by Jukes-Cantor distance model. Finally, our evolutionary trees of all modules was constructed and visualized by MEGA-CC[27] (version 11.0.11) and ggtree[28].

### GRINS analysis of AZL PKS
A sliding window with a step of 30 nt was used to split the AZL PKS gene cluster into 150 nt long fragments. Biopython pairwise2 module was applied to perform pairwise alignments between all non-overlapping 150 nt long fragments, and the highest value of DNA sequence identity was kept in every fragment. Regions at least 500 bp with the >80% average DNA sequence identity over five adjacent windows were specified as regions with high sequence identity. (G-C)/(G + C) and (T-A)/(T + A) are used to calculate the GC skew and TA skew, respectively. Finally, the region of GRINS containing the mean absolute GC and TA skew intensities >0.15 was identified among AZL PKS gene cluster[30].

### Construction of plasmids for in-frame deletion mutants $\Delta azlB$ and $\Delta azlB$ ($\Delta KT_6/KT_3$)
To construct the in-frame deletion mutants $\Delta azlB$ in vivo, firstly, two homologous fragments flanking *azlB* were amplified from *Streptomyces* sp. 211726 genome DNA by two pairs of primers $\Delta azlB$-L1 and $\Delta azlB$-L2, $\Delta azlB$-R1 and $\Delta azlB$-R2. Then, the *Streptomyces-E. coli* shuttle vector pYH7 digested by *Hin*dIII and *Nde*I were fused with the PCR products to conduct pWHU5037 by Gibson assembly. And similar procedures are performed to obtain pWHU5040 (Supplementary Table 6) with two pairs of primers, namely $KT_3$-sw-L1, $KT_3$-sw-L2, $KT_3$-sw-R1, and $KT_3$-sw-R2. Restriction endonuclease digestion and sequencing were applied to confirm the constructed plasmid.

### Construction of plasmids for domain replacement mutants in $\Delta azlB$
To construct the domain replacement mutants $\Delta azlB(\Delta KS_6/KS_3)$ in vivo, firstly, three fragments including $KS_3$ sequence and two homologous arms induced by three pairs of primers $KS_3$-sw-F, $KS_3$-sw-R, $KS_3$-sw-L1, $KS_3$-sw-L2, $KS_3$-sw-R1, and $KS_3$-sw-R2 were amplified from *Streptomyces* sp. 211726 and engineered $\Delta azlB$ genome DNA, respectively. Then, the *Streptomyces-E. coli* shuttle vector pYH7 digested by *Hin*dIII and *Nde*I were fused with the PCR products to conduct pWHU5038 by Gibson assembly. And similar procedures are performed to obtain pWHU5039, pWHU5041, pWHU5042, pWHU5043, pWHU5044 (Supplementary Table 6) with pairs of corresponding primers (Supplementary Table 3). Restriction endonuclease digestion and sequencing were applied to confirm the constructed plasmid.

## Procedure for construction and confirmation of mutants

The function-specific plasmids (Supplementary Table 6), conducted by the above methods, were introduced into *Streptomyces* sp. 211726 and engineered ΔazlB by conjugation with donor strain ET12567/pUZ8002 on SFM solid plates (with 120 mM $CaCl_2$). After incubating at 28 °C for 16 h, a mixture of apramycin (75 μg mL⁻¹) and nalidixic acid (30 μg mL⁻¹) in 1 mL were used to overlay each plate, which was then cultured at 28 °C for 5–6 days until the ex-conjugants emerged. To confirm the antibiotic resistance of the single exconjugants grown in this plate, they were patched on SFM solid medium plate with apramycin (25 μg mL⁻¹) and nalidixic acid (25 μg mL⁻¹) cultured at 28 °C for 3 or 4 days. Then verified ones were transferred on a fresh SFM plate without antibiotics by plate streaking to obtain single colonies. To screen for the double crossover mutant, those single clones were patched onto SFM plates with and without 25 μg mL⁻¹ apramycin, respectively. Total DNA isolation of candidate mutants with apramycin-sensitive phenotype was checked by PCR using one or two pairs of checking primers, such as ΔazlB-CK-F and ΔazlB-CK-R for screening of ΔazlB, $ER_{15}$-sw-CK-L1, $ER_{15}$-sw-CK-L2, $ER_{15}$-sw-CK-R1, and $ER_{15}$-sw-CK-R2 for screening of $ΔER_{1/2}/ER_{15}$, of which the PCR products were further confirmed by sequencing.

## Fermentation, isolation, and detection of AZL and derivatives

To obtain seed medium of *Streptomyces* sp. 211726 or mutant, a patch (~1 × 1 cm²) of these strains on SFM solid plate was inoculated into 20 mL TSBY liquid broth in 100 mL shake flask incubated at 28 °C with shaking at 220 rpm for 2 or 3 days. Then one percent of seed mycelium was inoculated into liquid SFM broth to ferment at 28 °C for 5–7 days at 220 rpm. To yield the crude extract of these strains, the fermentation products were centrifuged at 4696 × *g* for 15 min. The mycelium was then extracted with an identical volume of methanol two times, of which the supernatant was evaporated under reduced pressure after centrifugation. Subsequently, the concentrated products were dissolved in methanol again and respectively subjected to HPLC and LC-ESI-HRMS with totally different elution programs after filtration through a 0.22 μm Nylon66 membrane. AZL and related derivatives were detected by HPLC using a Phenomenex Luna C18 column (5 μm, 250 × 4.6 mm) at a flow rate of 1.0 mL min⁻¹ using a mobile phase of (A) $H_2O$ and (B) Acetonitrile. The separation gradient is as follows: 0–20 min, 35–65% B; 20–28 min, 65–95% B; 28–30 min, 95% B; 30–33 min, 95%-35% B; 33–38 min, 35% B. Detection by LC-ESI-HRMS was on a Thermo Electron LC-ESI-HRMS using positive mode electrospray ionization fitted with a Phenomenex Luna C18 column (5 μm, 250 × 4.6 mm) at a flow rate of 0.6 mL min⁻¹ using a mobile phase of (B) $H_2O$ and (C) Methanol. The separation gradient as follows: 0-2 min, 90%-20% B; 2–10 min, 20%-5% B; 10–11 min, 5%B; 11–12 min, 5–90%B; 2–15 min, 90%B. The mass spectrometer was set to full scan (from 200 to 2000 *m/z*).

## Large-scale fermentation and purification of AZL derivatives

Large-scale fermentations were conducted to characterize the structure of derivatives by NMR spectroscopy. Seed cultures of $ΔER_{1/2}/ER_{15}$, $ΔER_{1/2}/pol$-$ER_{1/2}$, and ΔazlB($ΔKA_6/KA_3$) were used to inoculate 10 L SFM liquid medium to obtain compound **1**, compound **2**, compound **3**, and **4**, respectively, which were developed at 28 °C with shaking at 220 rpm for five days and of which the crude products were extracted according to the extraction methods described above. Subsequently, they were subjected to chromatography on Sephadex LH-20 (40–70 μm) and eluted with a gradient concentration of acetonitrile-water (10–100%). Based on the analysis of HPLC and LC-ESI-HRMS, fractions containing target compounds were combined, and further purified by semipreparative HPLC (SHIMADZU, SPD-20A 230 V) with a Phenomenex Luna C18 column reverse phase column (5 μm, 250 × 10 mm) and eluted with proper methanol to yield corresponding target compounds.

## Characterization of AZL derivatives by NMR

1D (¹H, ¹³C, and DEPT) and 2D (¹H-¹H COSY, HSQC, HMBC, and ROESY) NMR spectra were collected on an Agilent-NMR-VNMRS 600 spectrometer (Supplementary Tables 1–4). Chemical shifts were reported in ppm using tetramethylsilane as an internal reference, and coupling constants were reported in Hz. NMR data processing was performed using MestreNova software. The molecular formula of compounds **1**–**4** were determined according to their LC-ESI-HRMS data. Comparative analysis of their ¹H (Supplementary Fig. 10, 16, 22, and 28) and ¹³C NMR data (Supplementary Fig. 11, 17, 23, and 29) revealed that most of their signals were almost identical, except that several signals shifted (as in compounds **1** and **2**) (Supplementary Figs. 11 and 17) or some signals disappeared (as in compounds **3** and **4**) (Supplementary Figs. 23 and 29). Through comprehensive analysis of their HSQC (Supplementary Figs. 12, 18, 24, and 30), ¹H-¹H COSY (Supplementary Figs. 13, 19, 25, and 31), HMBC (Supplementary Figs. 14, 20, 26, and 32), and ROESY (Supplementary Figs. 15, 21, 27, and 33) data, the chemical structures were constructed unambiguously.

## Reporting summary

Further information on research design is available in the Nature Portfolio Reporting Summary linked to this article.

## Data availability

The data generated in this study are provided in the Supplementary Information and Source Data files provided with this paper. Further data, plasmids and primers are also available from the corresponding author upon request. Source data are provided with this paper.

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

## Acknowledgements

This work was supported by National Key R&D Program of China (2018YFA0903200), the Funds for International Cooperation and Exchange of the National Natural Science Foundation of China (31920103001) and the National Natural Science Foundation of China (31270025) to Yu. S. We thank Prof. Zhaoyong Yang for providing the polaramycin producer strain *Streptomyces hygroscopicus* LP-93 as a gift, and Thomas J. Simpson for helpful discussion and proof-reading.

## Author contributions

G.Z. designed experiments and wrote the manuscript. Y.Z. designed and constructed mutants in this work. G.S. identified the structures of AZL derivatives. Fa.Z. and C.D. performed the phylogenetic analysis of AZL PKS. Ya.S. performed protein modeling. Z.H. performed the fermentation of mutants. K. H. provided the wild-type strain *Streptomyces* sp. 211726. Fu.Z. participated in discussion. P.F.L and Z.D. analyzed the data and revised the manuscript. Yu.S. conceived the overall project, analyzed the data, and wrote the manuscript.

## Competing interests

The authors declare no competing interests.
