## [Peer Review File · Nature Communications]

Insights into azalomycin F assembly-line contribute to evolution-guided polyketide synthase engineering and identification of intermodular recognitionREVIEWER COMMENTS

Reviewer #1 (Remarks to the Author):

In this study, the authors utilized elegantly designed genome editing experiments to manipulate the azalomycin F PKS assembly line in a natural producer in order to identify the domains responsible for an unusual trans-module acting ER. Gene deletion and domain substitution mutants were guided by a phylogenetic analysis of the PKS modules and domains, which suggested gene duplication and/or conversions between modules 3 and 6 could be responsible for their apparent relatedness. As a result of these PKS engineering experiments, four new azalomycin molecules were fully characterized, which differed in the extent of reduction and the macrocycle ring-size from the azalomycin F3a parent compound.

Although the experiments are well designed and executed, generally established methods were applied in the study. Engineering experiments were limited to swapping domains/modules having greater than 90% sequence identity in a single PKS system, which gave nice insights into the unusual cross-module ER activity but are of limited general interest due to the narrow scope. Comparing, for example, to Su et al. Engineering the stambomycin modular polyketide synthase yields 37-membered mini-stambomycins. *Nat Commun* 13, 515 (2022) as the standard of paper published in *Nature Communications*, the results presented in the current manuscript would be more appropriate to a specialized journal.

At this level of journal, I would have liked to see some follow-up in vitro work or at least mutagenesis at the subdomain level to characterize the interaction between ER1/2 and KS3 shown to be responsible for the trans-ER activity. Could methodologies like AlphaFold structure predictions provide any insights when comparing ER1/2 to ER15 or KS3 to KS6 to provide insights into how ER1/2 could interact with KS3? The bioactivity of azalomycins were not mentioned in the text but have been shown to bind phospholipids and disrupt membranes (<https://doi.org/10.1016/j.biopha.2018.11.067>). Perhaps the authors could have tested the bioactivity of the new congeners to add another dimension?

Other comments:

The authors should avoid the use of 'proved' in the context of interpreting their results as indisputable facts. Rather results are in support of stated hypotheses. Use of terminology such as 'the data showed/demonstrated/supported/indicated...' is recommended as an alternative. For example: page 14 '...but also proved evidence that the recognition of the docking domain between modules 1/2 and 3 does not directly participate in the process...' and page 20 '...we proved that KS3 is the anchor for ER1/2 leading to the catalysis of cross-module enoylreduction...'

In Figure 2b, the color scheme doesn't match parts a and c. compounds 1 should be labeled in a mustard yellow.

In Figure 3b, the GRINS analysis is extremely difficult to interpret. The caption is too minimalistic. The data is very condensed horizontally. The blue line indicating GC skew cannot be seen under the blue shading. AzlA was excluded from the GRINS analysis without explanation. This figure would be much more impactful if the results were spread over two stacked plots and explained more clearly, or if only a relevant portion was shown in the main text with the full plot in the SI.

A new result is introduced in the discussion section for the first time, 'to confirm the catalytic properties of pol-ER1/2...analysis of fermentation product analysis...' This result should be moved to the results section.

In the methods section, the wrong reference is given for GRINS analysis. It should be ref 29 (not 28).

Some terminology is a bit confusing or unusual and should be rephrased for clarity:

page 9. '...modules 3 and 6 were located on two branches with a clade connecting them...' could be stated as '...modules 3 and 6 were located on two branches of a single clade...'
page 19. '...the native interfaces between ACP1/2-KS3 and ACP6-KS7 was persevered...' instead recommend '...were maintained...'

Reviewer #2 (Remarks to the Author):

Wow! This paper is fantastic and really makes you think about the mechanistic enzymology and evolution of PKSs. The authors demonstrate through domain replacement that the ER that is harbored in the first (iterative) module of the azolomycin F PKS is indeed responsible for cross-module enoyl reduction in module 6. Through domain replacement including with a homologous domain in the polaromycin ER they indicate that other ER domains are not able to do this unique cross-talk however a homologous domain from poromycin is. The authors also demonstrate quite rigorously that interactions with the ketosynthase domain are critical for enabling this unusual cross talk. Interestingly, in modules 3 and 6, the KR domain has a truncation which perhaps allows for more spatial access.

The functional data, the rationale for creating mutants, and the phylogenetic analysis is all quite strong and their argument is well made. The presence of a similar GRINS sequence in modules 3 and 6 also is also compelling for their evolutionary story. I highly recommend publishing this high quality work in Nature Communications.

I do however have a couple suggestions to improve the manuscript.

1. I feel like Figure 6, the cartoon could be drawn in a more informative way. We actually know quite a lot about the shape of these domains (even if the module structure is still unknown and there are still some disagreement with what has been observed in Cryo-EM structures versus in multi-domain structures through crystallography or through SAXS in solution phase or through recent antibody bound structures determined by Khosla). Additionally, there are compelling models put forth by Keatinge-Clay (Zheng et. al, 2012, Nat. Chem. Bio., 8, 615) and Khosla (Klaus et. al, JACS Au, 2021 2162-2171; Cogan et. al, Science, 2021, 374, 729). Given how critical spatial fit is to understanding the module, I think it would be helpful to see this depicted as a cartoon that better represented domain structures and relative size. Of course without structural information supporting it, it would remain just a model, but I think there is a better way to capture all the information presented in this article. The KR domain having additional space is not depicted at all. It is hard for me to visualize the interaction between the KS and the ER given what we know about KS-AT structures. It need not be exact, but it would be more useful to understand the model than the full cartoon version as drawn.

2. I would also like to see discussion of the KR truncation. PKS KR domains consist of two Rossmann folds, one that does not bind NADPH, often called the structural subdomain. From the sequence alignment in supplemental figure 3, it appears that this is present in part of the structural subdomain which occurs N-terminally. But does it disrupt the Rossmann fold entirely? The degree to which it does or does not should be fairly clear from homology modeling. Exactly how much space does this give, if there is a truncated Rossmann fold?

Reviewer #3 (Remarks to the Author):

The main results of this work are 1) that the cross-module ER domain in azolomycin F biosynthesis can be replaced by other ER domains both from within the PKS and from other PKSs, and only one of these (one that has this function in another system) can catalyse significant cross-module enoyl reduction. 2) Modules 3 and 6 from this PKS are evolutionarily related. 3) Intermediates can be transferred directly from module 1/2 to module 6 but no enoyl reduction occurs. 4) The KS domain is the key determinant of enoyl reduction activity across modules and incorporation of the module 3 KS domain into module 6 reinstates cross-module reduction.

Points 1 to 3 although interesting are somewhat incremental advances on previous work, but point 4 represents a noteworthy finding that will be of significance to the field

In general the work supports the conclusions and aims, some comments below.

1) The experiment that is missing in the dataset in figure 2b, and should be included, is an ER1/2 domain KO, without complementation. If this abolishes production entirely it would suggest KS3 is very selective for the reduced intermediate. Also in figure 2b it is impossible to tell which strains actually produce 2, nor is this explained well in the text. Eg is it produced in the WT strain? The authors don't really provide a good explanation for why the C40-41 bond is reduced in all the mutants when it is not in the WT strain. Perhaps the swapped in ER domains are just quicker, so always catalyse reduction before chain extension can occur?

2) Not clear why the discussion of fermentation of *S. hydroscopicus* LP-93 is in the discussion section rather than the results, it would be better discussed with the above. The use of the phrase 'toggled off' is odd, as surely this is just the kinetics favouring reduction or chain extension. Similarly use of 'inactive' here is misleading. Furthermore, towards the end of this paragraph whether enoyl reduction happens in module 1 or 2 is assumed to be due to substrate preference, it could simply be that KS3 only accepts reduced substrates so the elongated ACP2 tethered substrates sits longer and is more easily reduced.

3) There is a discrepancy between the authors previous in vitro work on this system and this publication. Their previous work suggested the KR domain is essential for cross-module enoyl reduction. Here they suggest it is not. However, here they swap the KR1/2 domain rather than delete it. The most likely explanation is therefore that the KR15 domain can successfully complement whatever the KR1/2 domain is doing to facilitate transfer. A simple way to test this, which should be done, would be to replicate this experiment in the in vitro system the authors have previously published.

4) The comment on page 16 that the absence of compound 4 in the Δ azlB mutant suggests the docking domain is not directly involved in cross module enoyl reduction is not correct. It shows that it cannot be the ONLY thing involved but not that it is not involved at all, as it is present in all of the experiments.

Some minor details

Page 3 – the authors claim that with four types of module, a 6 module PKS would be able to yield more than 100,000 different polyketides. That maths doesn't really make sense. It would be 4^6 so 4096. If they mean because of the various permutations of malonyl/methylmalonyl/S or R KR specificity etc this needs to be explicitly stated.

Supplementary figure 2 shows the ER domain in module 1 blacked out for PolA, but presumably it is active in this system.

In general I found the manuscript quite hard to follow. It could do with a thorough proof read and the sentence structure simplified, to make it more accessible to a broad audience.

I am not clear what 'non-specific cross module enoyl reduction' is supposed to be. Again – do the authors just mean whether it happens quickly or slowly or are they suggesting it is done by an ER domain separate to the PKS?

The methodology and description of the methodology is sound.

Response to reviewer 1

General comments

Q1: At this level of journal, I would have liked to see some follow-up in vitro work or at least mutagenesis at the subdomain level to characterize the interaction between ER_{1/2} and KS₃ shown to be responsible for the trans-ER activity. Could methodologies like AlphaFold structure predictions provide any insights when comparing ER_{1/2} to ER₁₅ or KS₃ to KS₆ to provide insights into how ER_{1/2} could interact with KS₃?

A1: Thank you for your constructive suggestion. This molecular basis for ER_{1/2} and KS₃ interaction is a fascinating question that we also very much hope to solve. Except for a lot of informatic analysis and genetic investigations, we have indeed performed 3D structure modeling of ER_{1/2}, ER₁₅, KS₃, and KS₆ domains using AlphaFold (Jumper, J. *et al. Nature* 2021, 596, 583–589), and conducted protein-protein docking analysis (Kozakov, D. *et al. Nat. Protoc.* 2017, 12, 255-278). The structure alignment results showed no obvious structural difference between ER_{1/2} and ER₁₅, except for the slight difference (**DDH** motif in ER_{1/2} verse **GEA** in ER₁₅) in the substrate binding loop (Khare, D. *et al. Structure* 2015, 23, 2213-2223, Figure 1a here). Considering that a similar conserved **LDD** motif in B-type KR domain determines the direction of the substrate entering the reaction pocket (Liu, C. *et al. J. Struct. Biol.* 2018, 203, 135-141), we thus speculated that the **DDH** motif could be correlated with the unusual catalytic behavior of ER_{1/2}. To test this, we mutated **DDH** into **AAA** (Figure 1b here). However, the fermentation product of this mutant showed no detectable difference compared with the wild-type. Meanwhile, we also carefully compared the structure between KS₃ and KS₆. As shown in Figure 1c here, the modeling 3D structure of KS₃ is almost identical to that of KS₆. Given this, we propose that certain unidentified amino acids that do not lead to an apparent structural difference in ER_{1/2} and KS₃ might be responsible for the cross-module enoylreduction.

Figure 1. AlphaFold modeling structure comparison and site-directed mutation. (a) The structural alignment between ER_{1/2} (brown) and ER₁₅ (cyan). The obvious structural difference is colored in red in ER_{1/2}. (b) Site-directed mutation of ER_{1/2} in vivo. The structural difference of ER_{1/2} was mutated by changing the sequence DDH into AAA in vivo. The mutation was confirmed by PCR and sequencing. (c) The structural alignment between KS₃ (pink) and KS₆ (pale lavender).

To identify the residues or sub-structures relevant to the protein-protein interactions, we performed the alignment of multiple ER domains, which are homologs of AZL ER_{1/2} and ER₁₅ from the other AZL and related analog producer (Figure 2, here). The alignment results combined with ER_{1/2} modeling structure showed that the residue Y30, Y265 around the substrate tunnel, as well as P170 and Y258 around the cofactor NADPH were highly conserved in AZL ER_{1/2} and homologs (Figures 2 and 3a, here). Meanwhile, an extra conserved motif (YDLVVADYDWWQR) termed as motif 1 here around the substrate tunnel was also found in AZL ER_{1/2} and almost homologs. Besides, there was a conserved I126 which might be important in interaction with motif

1. Considering ER_{1/2} could not be complemented by ER₁₅, to explore the function of these conserved residues and motif, we therefore mutated the mentioned residues and motif into the corresponding residues and motif in ER₁₅ to construct mutants ER_{1/2}(Y30E), ER_{1/2}(I126Y), ER_{1/2}(P170E), ER_{1/2}(Y258F), ER_{1/2}(Y265P), ER_{1/2}-motif1-sw (Figures 3b-g, here). And we also constructed the mutants in which the mentioned residues were mutated into alanine. Unfortunately, no detectable difference was found compared with the wild-type strain, indicating that other residues far away from the substrate binding pocket maybe the candidates.

Figure 2. The multiple sequence alignment of ER domains. These ER domains are from the AZL and related analog PKS assembly-line except for rap-ER₇. The red asterisks indicate the conserved and studied residues in ER_{1/2} domains. AZL ER_{1/2} and homologs are outlined in red. Tu4113-ER_{1/2}/ER₁₅ (*Streptomyces violaceusniger* Tu

4113, NC_015957.1), M2017417-ER_{1/2}/ER₁₅ (*Streptomyces spectabilis* CCTCC M2017417, KY593296), DSM4137-ER_{1/2}/ER₁₅ (*Streptomyces* sp. DSM4137, CP023992.1), IMB7-145-ER_{1/2}/ER₁₅ (*Streptomyces* sp. IMB7-145, MF671979), 211726-ER_{1/2}/ER₁₅ (*Streptomyces* sp. 211726, KY484834), poI-ER_{1/2}/ER₁₅ (*Streptomyces hygrosopicus* LP-93), rap-ER₇ (*Streptomyces hygrosopicus* LP-93).

Figure 3. The confirmation of the conserved residues in ER_{1/2}. (a) The distribution of these conserved residues in 3D modeling structure of ER_{1/2}. Construction and confirmation of ER_{1/2}(Y30E) (b), ER_{1/2}(I126Y) (c), ER_{1/2}(P170E) (d), ER_{1/2}(Y258F) (e), ER_{1/2}(Y265P) (f) and ER_{1/2}-motif1-sw (g).

To gain more information about the interaction between ER_{1/2} and KS₃, ClusPro was used to construct the possible docking model (Kozakov, D. *et al. Nat. Protoc.* 2017, 12, 255-278). Considering the potential impact of AT and KR on KS and ER domains, we performed four groups of docking experiments using ER_{1/2}-KR_{1/2} with KS₃-AT₃, ER_{1/2}-KR_{1/2} with KS₆-AT₆, ER₁₅-KR₁₅ with KS₃-AT₃, and ER₁₅-KR₁₅ with KS₆-AT₆, respectively. The docking models that ER_{1/2} docked with KS₃ or KS₆ were obtained using the hydrophobic-favored algorithm, but no significant difference between these two models. However, in all obtained docking models, there was no such model that ER₁₅ docked with KS₃ or KS₆, while ER₁₅ just docked with AT₆. Analysis of the docking models revealed that the docked KS₃ and KS₆ domains blocked the substrate tunnel of ER_{1/2}, and the dimeric interface of KS₃ was also occupied (Figure 4 here). Accordingly, we thought this docking model was not conceivable and did not reflect the real conformation in the cross-module enoylreduction state of ER_{1/2} and KS₃. Subsequently, we constructed the dimeric KS₃-AT₃ and KS₆-AT₆ domains using AlphaFold-multimer to perform the above docking experiments again. However, no reliable docking model was obtained. Considering the dynamic change of the module conformation during the different reaction processes, it is not credible to construct the docking model with the structure in a specific conformation. It is known that a vast challenge exists in characterizing the structure of cis-AT PKS module, and there is no fully reduced module structure has been reported so far. We have tried our best to investigate the interaction between ER_{1/2} and KS₃. However, it is tough to reveal the mechanism without the structural information. We hope to capture the complex structure of module 1/2 and module 3 using Cryo-EM, and to provide more detailed insights into how ER_{1/2} interacts with KS₃ in the future.

Figure 4. Modeling structure and docking model of ER_{1/2}-KR_{1/2} and KS₃-AT₃. (a) Molecular surface of modeling structure of ER_{1/2}-KR_{1/2}, and 60° rotation anti-clockwise along the Y axis to (b). (c) Docking model of ER_{1/2}-KR_{1/2} with KS₃-AT₃.

Q2: The bioactivity of azalomycins were not mentioned in the text but have been shown to bind phospholipids and disrupt membranes (<https://doi.org/10.1016/j.biopha.2018.11.067>). Perhaps the authors could have tested the bioactivity of the new congeners to add another dimension?

A2: Yes, we have actually tested the bioactivity of AZL new analogs to anti-MRSA, anti-*Candida albicans* and anti-cancer (myeloma, myeloid leukemia, etc.). Unfortunately, the results showed no better or even worse than AZL, which might be caused by the unstable combination with the target due to the shrunken ring.

Minors:

Q3: The authors should avoid the use of 'proved' in the context of interpreting their results as indisputable facts. Rather results are in support of stated hypotheses. Use of terminology such as 'the data

showed/demonstrated/supported/indicated...' is recommended as an alternative. For example: page 14 '...but also proved evidence that the recognition of the docking domain between modules 1/2 and 3 does not directly participate in the process...' and page 20 '...we proved that KS₃ is the anchor for ER_{1/2} leading to the catalysis of cross-module enoylreduction...'

A3: We have revised our manuscript as your suggested.

Q4: In Figure 2b, the color scheme doesn't match parts a and c. compounds 1 should be labeled in a mustard yellow.

A4: We have amended it in the revised manuscript.

Q5: In Figure 3b, the GRINS analysis is extremely difficult to interpret. The caption is too minimalistic. The data is very condensed horizontally. The blue line indicating GC skew cannot be seen under the blue shading. AzIA was excluded from the GRINS analysis without explanation. This figure would be much more impactful if the results were spread over two stacked plots and explained more clearly, or if only a relevant portion was shown in the main text with the full plot in the SI.

A5: In the natural organization of AZL biosynthetic gene cluster in the chromosome, the starter *azIA* is unusually located at the end of AZL PKS cluster (*azIB-E*, *azIA*, Fig. 3 in manuscript), instead of intuitively existing upstream of *azIB* as it is in the proposed assembly-line (*azIA-E*, Fig. 1 in manuscript). It would be easy to miss the right-most AzIA in Fig. 3b in manuscript.

Nivina and coworkers found intense GC and TA skew in most duplicated regions on the PKS assembly-line. Meanwhile, they found that paralogous modules containing GC and TA skews are often more similar to each other than orthologous ones. Considering sequences with G+C bias are known to recombine more frequently, they proposed that gene conversion, which might be caused by their reduced sequence complexity, could be the reason for the

more similarity between paralogous modules. Accordingly, they proposed that these intense nucleotide skews would play a role in accelerating the diversification of closely related biosynthetic clusters, and they designated these genetic elements as GRINS that as long (> 500 bp) duplicated regions (> 80% DNA sequence identity to another region within the same PKS) of intense nucleotide skews (means of absolute GC and TA skew values within the region > 0.15). According to the definition, GC and TA skew are highly relevant parameters, so it would be better to show these parameters in one plot. Additionally, the GRINS presentation of the whole AZL PKS assembly-line in one plot would comparatively understand the distribution of GRINS. To make GC skew looks more apparent, we have adjusted its contrast. Meanwhile, we also enriched the figure legend for better interpretation in the revised manuscript.

Q6: A new result is introduced in the discussion section for the first time, 'to confirm the catalytic properties of pol-ER_{1/2}...analysis of fermentation product analysis...' This result should be moved to the results section.

A6: As suggested, we have moved to the result section in the revised version.

Q7: In the methods section, the wrong reference is given for GRINS analysis. It should be ref 29 (not 28).

A7: We have corrected it in the revised version.

Q8: Some terminology is a bit confusing or unusual and should be rephrased for clarity: page 9. '...modules 3 and 6 were located on two branches with a clade connecting them...' could be stated as '...modules 3 and 6 were located on two branches of a single clade...' page 19. '...the native interfaces between ACP1/2-KS3 and ACP6-KS7 was persevered...' instead recommend '...were maintained...'

A8: We have revised it as suggested.

Response to reviewer 2

Q1: I feel like Figure 6, the cartoon could be drawn in a more informative way. We actually know quite a lot about the shape of these domains (even if the module structure is still unknown and there are still some disagreements with what has been observed in Cryo-EM structures versus in multi-domain structures through crystallography or through SAXS in solution phase or through recent antibody bound structures determined by Khosla). Additionally, there are compelling models put forth by Keatinge-Clay (Zheng et. al, 2012, Nat. Chem. Bio., 8, 615) and Khosla (Klaus et. al, JACS, 2021 2162-2171; Cogan et. al, Science, 2021, 374, 729). Given how critical spatial fit is to understanding the module, I think it would be helpful to see this depicted as a cartoon that better represented domain structures and relative size. Of course, without structural information supporting it, it would remain just a model, but I think there is a better way to capture all the information presented in this article. The KR domain having additional space is not depicted at all. It is hard for me to visualize the interaction between the KS and the ER given what we know about KS-AT structures. It need not be exact, but it would be more useful to understand the model than the full cartoon version as drawn.

A1: Thank you for your very helpful suggestion. As there is no information about the structure of AZL module 1/2 and module 3, we can only draw a cartoon according to the current structural information and our understanding to show the structure topologies of module 1/2 and module 3 during the cross-module enoylreduction state. Based on the turnstile-closed structure of AT in DEBS M1 (Cogan, D. P. *et al. Science* 2021, **374**, 729-734), we positioned AT₃ in a flexed conformation, as shown in Figure 5 here. Meanwhile, the AlphaFold modeling structure showed that the conformation of DH and KR domains were similar to the DH and KR domains of mammalian fatty acid and lovastatin synthase

(Maier, T. *et al. Science* 2008, 321, 1315-1322; Wang, J. *et al. Nat. Commun.* 2021, 12, 867). Additionally, in the light of the structural information of CTB1 SAT-KS-MAT in which the starter unit acyltransferases (SAT) connected with the KS domain and the active site entrance of each SAT points into the reaction chamber of CTB1 (Herbst, D. A. *et al. Nat. Chem. Biol.* 2018, 14, 474-479). Similarly, we speculate that the conformation of $KR_{1/2}$ and $ER_{1/2}$ would undergo a certain angle of rotation along the X axis resulting in the connection between $ER_{1/2}$ and KS_3 with the substrate tunnel orientation of $ER_{1/2}$ toward the reaction chamber of module 3 in the process of cross-module enoylreduction (Figure 5 here). Therefore, we have drawn a full cartoon version as suggested and added it as Supplementary Fig. 8. in the revised manuscript, together with Fig. 6, to help better understand our model.

Figure 5. The proposed domain topologies of module 1/2 and module 3 in the state of cross-module enoylreduction. In the process of cross-module enoylreduction, a certain angle of clockwise rotation along the X axis of $KR_{1/2}$ and $ER_{1/2}$ facilitated the connection between $ER_{1/2}$ and KS_3 with the substrate tunnel orientation of $ER_{1/2}$ toward the reaction chamber of module 3, which made it accessible for the intermediate tethered onto ACP_3 to be reduced.

Q2: I would also like to see discussion of the KR truncation. PKS KR domains consist of two Rossmann folds, one that does not bind NADPH, often called the structural subdomain. From the sequence alignment in supplemental figure 3, it appears that this is present in part of the structural subdomain which occurs N-terminally. But does it disrupt the Rossmann fold entirely? The degree to which it does or does not should be fairly clear from homology modeling. Exactly how much space does this give, if there is a truncated Rossmann fold?

A2: Thanks for your inspiring question. To scrutinize the difference among truncated KR₃, KR₆, and other canonical KR domains such as KR₅, we performed the 3D structure prediction of them using AlphaFold (Jumper, J. *et al. Nature* 2021, 596, 583–589). Carefully comparison of these KR domains revealed an obvious structural difference in truncated KR domains, in which two α -helixes and two β -folds are missing in the structural subdomain. Rossmann fold consists of two cassettes containing three parallel β -fold and two pairs of α -helix. The 3D structure modeling results showed that the missing α -helix and β -fold only disrupt the Rossmann fold but do not affect the folding of the structural subdomain, as they lie on the outmost side (Figure 6 here). Results from chimera show that the volume of KR₃, KR₆, and KR₅ (Pettersen, E. F. *et al. J. Comput. Chem.* 2004, 25, 1605-1612) are 30.45 e³, 30.94 e³, and 33.22 e³, respectively. The calculation results revealed that the truncated structure might provide other domains in module 3 and module 6 additional two e³ space. So, we have added the following Figure 6 in the revised Supplementary Information as Supplementary Fig. 9.

Figure 6. The structural alignment of KR domains. (a) The structural alignment between KR₃ (pale purple), and KR₆ (light cyan) show a highly structural coincidence. (b) The structural alignment among KR₃ (pale purple), KR₆ (light cyan), and KR₅ (yellow) reveal a truncated structural subdomain in KR₃ and KR₆, where two α -helixes and two β -folds (red) existing only in KR₅ are missing. These KR structures are obtained using AlphaFold.

Response to reviewer 3

Majors

Q1: The experiment that is missing in the dataset in figure 2b, and should be included, is an ER_{1/2} domain KO, without complementation. If this abolishes production entirely it would suggest KS₃ is very selective for the reduced intermediate.

A1: Thank you for your suggestion. To be avoided disturbing the native conformation of the PKS module, it is more suitable and general to inactivate domains using site-directed mutagenesis to investigate the function of domains. Hence, in our previous work (Zhai, G. *et al.*. *Angew. Chem. Int. Ed.* 2020, 56, 22738-22742), we mutated the NADPH binding site GGVG into AAVA and SPVG of ER_{1/2} in vivo and in vitro respectively (Figure S1 and S24 in Zhai, G. *et al.* *Angew. Chem. Int. Ed.* 2020, 56, 22738-22742), rather than knocking it out, to explore its catalytic behavior. The in vivo result showed that the mutation of ER_{1/2} domain entirely abolished the production of AZL while analog bearing three conjugated double bonds, which correspond to the product of the first three rounds of extension, was accumulated with an extremely low production

level (Figure 2 in Zhai, G. *et al. Angew. Chem. Int. Ed.* 2020, 56, 22738-22742). Moreover, the in vitro result also showed that mutation of ER_{1/2} domain completely abolished the cross-module enoylreduction activity of module 1/2. These results together provide direct genetic and enzymatic evidence that ER_{1/2} is the necessary and sufficient catalyst for cross-module enoylreduction. Besides, these results confirmed that mutation of NADPH binding site completely inactivated the function of ER_{1/2} domain, which should be equivalent to the result of ER_{1/2} domain knock-out and could be used as the control of ER domain swapping mutants in this work. Taken above together, we constructed Δ ER_{1/2}/ER₁₅, Δ ER_{1/2}/rap-ER₇ and Δ ER_{1/2}/pol-ER_{1/2} via the direct replacement of ER_{1/2} with these ER domains rather than complementing them based on the ER_{1/2} domain knocking-out mutant. Although the inactivation of ER_{1/2} domain does not abolish the production of this mutated PKS assembly-line, it is rational to suppose that the strict substrate selectivity of KS₃ and KS₄ domains would be reasons for the extreme reduction of production level due to the slower transfer of structure changed substrate.

Q2: Also in figure 2b it is impossible to tell which strains actually produce 2, nor is this explained well in the text. Eg is it produced in the WT strain?

A2: We have added detailed description in figure legend in the revised manuscript. And compound 2 is also produced in the WT strain.

Q3: The authors don't really provide a good explanation for why the C40-41 bond is reduced in all the mutants when it is not in the WT strain. Perhaps the swapped in ER domains are just quicker, so always catalyse reduction before chain extension can occur?

A3: We proposed that these swapped ER domains have a broader range of substrate selectivity than ER_{1/2}. So, they can efficiently recognize and catalyze the nascent chain reduction during the first round of extension, resulting in the

saturated bond of C40-C41 before the next round of chain extension.

Q4: Not clear why the discussion of fermentation of *S. hydroscopicus* LP-93 is in the discussion section rather than the results, it would be better discussed with the above.

A4: We have moved this part of “While a significantand off-spring genes” to the Result section as suggested.

Q5: The use of the phrase ‘toggled off’ is odd, as surely this is just the kinetics favouring reduction or chain extension.

A5: For better understanding, we have modified this sentence as follows “..., suggesting that pol-ER1/2 was insufficient to catalyze the enoylreduction on all the nascent chain in the first round of chain elongation before translocation, which resulted in the accumulation of C40-C41 unsaturated product.” in the revised manuscript.

Q6: Similarly use of ‘inactive’ here is misleading. Furthermore, towards the end of this paragraph whether enoyl reduction happens in module 1 or 2 is assumed to be due to substrate preference, it could simply be that KS₃ only accepts reduced substrates so the elongated ACP₂ tethered substrates sits longer and is more easily reduced.

A6: We have changed ‘inactive’ to ‘not work’ in the revised manuscript. We agree that the substrate preference of KS₃ may be one of the reasons for the enoylreduction in the second round of chain elongation and for the different chain elongation outcomes between module 1 and module 2. Meanwhile, the saturated intermediate resulted from the replacement of azl-ER_{1/2} with pol-ER_{1/2} in the first round of chain extension suggested that the substrate preference of azl-ER_{1/2} and pol-ER_{1/2} is different, and the substrate preference of ER domain could also be the reason for the enoylreduction. Therefore, we discussed the

substrate preference of azl-ER_{1/2} and pol-ER_{1/2} during the first round of chain elongation at the end of the first section of the Results.

Q7: There is a discrepancy between the authors previous in vitro work on this system and this publication. Their previous work suggested the KR domain is essential for cross-module enoyl reduction. Here they suggest it is not. However, here they swap the KR_{1/2} domain rather than delete it. The most likely explanation is therefore that the KR₁₅ domain can successfully complement whatever the KR_{1/2} domain is doing to facilitate transfer. A simple way to test this, which should be done, would be to replicate this experiment in the in vitro system the authors have previously published.

A7: In our previous work, in vitro assay results showed that the ER_{1/2}-KR_{1/2} domain is the minimal cassette for cross-module enoylreduction (Figure 3 in Zhai, G. *et al. Angew. Chem. Int. Ed.* 2020, 56, 22738-22742), and the inactivation of the KR_{1/2} domain has no detectable effect on the cross-module enoylreduction of ER_{1/2} (Figure S26 in Zhai, G. *et al. Angew. Chem. Int. Ed.* 2020, 56, 22738-22742), suggesting that KR_{1/2} domain is important in preserving contacts with ER_{1/2} and stabilizing the ER_{1/2} in its native conformation (Zhai, G. *et al. Angew. Chem. Int. Ed.* 2020, 56, 22738-22742). In this work, the ER domain swapping experiments revealed a trace amount of cross-module enoylreduction product accumulated in these mutants, which demonstrated that swapped ER domains could also catalyze the cross-module enoylreduction with extremely low efficiency. One possibility could be that KR_{1/2} may interact with module 3 to facilitate the cross-module enoylreduction. According to our previous result (Figure 2, Zhai, G. *et al. Angew. Chem. Int. Ed.* 2020, 56, 22738-22742), the inactivation of ER_{1/2} completely abolished the cross-module enoylreduction on the intermediate of module 3, which suggested that ER₁₅-KR₁₅ didomain in AZL PKS assembly-line cannot interact with module 3; otherwise, there should be a certain amount of cross-module enoylreduction

product created. To test whether it is the reason that the trace of cross-module enoylreduction product results from the interaction between the KR_{1/2} domain and module 3 in mutant $\Delta ER_{1/2}/ER_{15}$, we, therefore, constructed the didomain replacement mutant $\Delta ER_{1/2}-KR_{1/2}/ER_{15}-KR_{15}$. Analysis of the fermentation product revealed a similar trace of cross-module enoylreduction product accumulated in $\Delta ER_{1/2}-KR_{1/2}/ER_{15}-KR_{15}$ compared with $\Delta ER_{1/2}/ER_{15}$. This result implies that there should be other factors, rather than the proposed interaction between KR_{1/2} and module 3 accounts for this extremely low cross-module enoylreduction. So, we think there is no conflict between our current results and previous work. To be avoided misleading by our previous description, we amended the sentence "..., which ruled out the possible involvement of KR_{1/2} in the cross-module enoylreduction" as "which ruled out the possible interaction between KR_{1/2} and module 3 in the cross-module enoylreduction".

Q8: The comment on page 16 that the absence of compound 4 in the $\Delta azIB$ mutant suggests the docking domain is not directly involved in cross module enoyl reduction is not correct. It shows that it cannot be the ONLY thing involved but not that it is not involved at all, as it is present in all of the experiments.

A8: Docking domains play important roles in module recognition and nascent chain transfer. We agreed that the specific docking domains between module 1/2 and module 3 assisted the cross-module enoylreduction in some of extent. Because of the in vitro assay and the product of $\Delta azIB$ showed that the docking domain is not essential for cross-module enoylreduction. So, we have changed 'not directly involved' to 'could not be the only thing' in the revised version.

Minors

Q9: Page 3 – the authors claim that with four types of module, a 6 module PKS would be able to yield more than 100,000 different polyketides. That maths

doesn't really make sense. It would be 46 so 4096. If they mean because of the various permutations of malonyl/methylmalonyl/S or R KR specificity etc this needs to be explicitly stated.

A9: We have added explicitly explanation in the revised manuscript as suggested.

Q10: Supplementary figure 2 shows the ER domain in module 1 blacked out for PolA, but presumably it is active in this system.

A10: We have corrected it in the revised Supplementary Fig. 2.

Q11: In general I found the manuscript quite hard to follow. It could do with a thorough proof read and the sentence structure simplified, to make it more accessible to a broad audience.

A11: We have refined our manuscript with the help of native speaker Prof. Tomas J. Simpson to make it more accessible to a broad audience.

Q12: I am not clear what 'non-specific cross module enoyl reduction' is supposed to be. Again – do the authors just mean whether it happens quickly or slowly or are they suggesting it is done by an ER domain separate to the PKS?

A12: Under normal conditions, the transfer of intermediates is so quick that the slower non-specific activities of domains do not have time to occur. While in the engineered PKS assembly-line, increased dwell time of intermediates on the synthetase due to the inauthentic substrate, allowing other catalytic properties of enzymes to be manifested (Wu, J. *et al. ChemBioChem* 2008, 9, 1500-1508; Taft, F. *et al. J. Am. Chem. Soc.* 2009, 131, 3812-3813; Yang, X. L. *et al. Chem. Sci.* 2019, 10, 8478-8489). Similarly, in our case, due to the transfer delay of the unreduced α,β -unsaturated nascent chain tethered onto ACP₃, allowing the other swapped ER domains to catalyze the reduction of this chain with

extremely low efficiency. So, we termed this as 'non-specific cross-module enoylreduction' in this study.

REVIEWER COMMENTS

Reviewer #1 (Remarks to the Author):

Thanks to the authors for their detailed rebuttal. I accept the changes to the manuscript and have no further concerns about the overall content. The authors could optionally include their bioactivity data for the new compounds (e.g. 3 and 4) compared to F3a, but I understand that there is limited space in communication length papers.

I was happy to see that the authors caught a few errors themselves, e.g. Brian et al  Lowry et al. on line 375). I suggest a few other minor edits:

line 57. 'The interaction of the docking domains on the C-terminal of the upstream module and the N-terminal of the adjacent downstream module guarantees the specific recognition between the corresponding modules' terminal should be changed to 'terminal end' or 'terminus'

line 118. 'fermentation product' should be plural '... products'

line 177. 'was not work during' should be 'did not function during' or 'was not working during'

line 181. 'second and third round extension, the preferences of the substrate during the first round of chain extension mainly were different' The substrate cannot have a preference. Rather the enzyme has a substrate preference and extensions should be plural. Rephrase to '...second and third round extensions, the substrate preferences during the first round of chain extension were mostly different...'

Line 198. 'Thus, Wang and coworkers proved that a discrete ER LovC binds with the AT domain in LovB, selectively catalyzing the enoyl reduction required at different stages of chain growth' suggest adding 'during lovastatin biosynthesis' to the end of the sentence to give context.

As I mentioned in the first round of reviews, I found the experiments well-executed, but had concerns about the limited general interest of the manuscript due to its narrow scope. These concerns do not appear to be shared by the other reviewers; therefore, I will defer to the majority judgement and the editor in this regard.

Reviewer #2 (Remarks to the Author):

All concerns with the manuscript have been adequately addressed.

Reviewer #3 (Remarks to the Author):

In the most part the authors have addressed my concerns.

The only thing I still think needs further clarity is the importance of the KR domain. If I now understand correctly, the authors are saying that since they were surprised that a small amount of enoyl reduction was observed in their ER15 swap they thought it might be because KR1/2 was facilitating it. When they then also swapped KR1/2 to KR15 this had no effect so ruled out this possibility. All OK but I don't think it rules out the possibility of KR1/2 being involved in facilitating the interaction between module 1/2 and module 3 in the native system, as is implied in the manuscript. It could be that ER15 can complement the activity of KR15 when inserted into module 1/2, it could be that KR1/2 can only interact with module 3 when ER1/2 is present. Perhaps these options are less likely than KR1/2 not being involved but the experiments done to date don't 'rule it out' as is stated here.

Response to reviewer 1

Q1: The authors could optionally include their bioactivity data for the new compounds (e.g. 3 and 4) compared to F3a, but I understand that there is limited space in communication length papers.

A1: We appreciate your constructive suggestions. Initially, we were also wondering about the bioactivity of the new compounds and tested them as previously reported. However, the results showed no better or even worse than AZL, which might be caused by the unstable combination with the target due to the shrunken ring, and limited space in paper length. We thus did not include this data in our manuscript.

Q2: line 57. 'The interaction of the docking domains on the C-terminal of the upstream module and the N-terminal of the adjacent downstream module guarantees the specific recognition between the corresponding modules' terminal should be changed to 'terminal end' or 'terminus'

A2: Thank you for careful review, we have changed 'terminal' to 'terminus'.

Q3: line 118. 'fermentation product' should be plural '... products'

A3: We have revised it as your suggestion.

Q4: line 177. 'was not work during' should be 'did not function during' or 'was not working during'

A4: We have changed 'was not work during' to 'was not working during'.

Q5: line 181. 'second and third round extension, the preferences of the substrate during the first round of chain extension mainly were different' The substrate cannot have a preference. Rather the enzyme has a substrate preference and extensions should be plural. Rephrase to '...second and third round extensions, the substrate preferences during the first round of chain

extension were mostly different...'

A5: We have revised it as your suggestion.

Q6: line 198. 'Thus, Wang and coworkers proved that a discrete ER LovC binds with the AT domain in LovB, selectively catalyzing the enoyl reduction required at different stages of chain growth' suggest adding 'during lovastatin biosynthesis' to the end of the sentence to give context.

A6: We have added 'during lovastatin biosynthesis' in the revised manuscript as your suggestion.

Response to reviewer 3

Q1: The only thing I still think needs further clarity is the importance of the KR domain. If I now understand correctly, the authors are saying that since they were surprised that a small amount of enoyl reduction was observed in their ER₁₅ swap they thought it might be because KR_{1/2} was facilitating it. When they then also swapped KR_{1/2} to KR₁₅ this had no effect so ruled out this possibility. All OK but I don't think it rules out the possibility of KR_{1/2} being involved in facilitating the interaction between module 1/2 and module 3 in the native system, as is implied in the manuscript. It could be that ER₁₅ can complement the activity of KR₁₅ when inserted into module 1/2, it could be that KR_{1/2} can only interact with module 3 when ER_{1/2} is present. Perhaps these options are less likely than KR_{1/2} not being involved but the experiments done to date don't 'rule it out' as is stated here.

A1: We apologize for the confusion and thank you for your helpful discussion. As the cryo-EM structures of the hybrid DEBS M3/1(KS₃-AT₃-KR₁-ACP₁) and DEBS M1(KS₁-AT₁-KR₁-ACP₁) together revealed that the complementary interaction between cognate domains within a module is quite important in stabilizing the conformation of the specific domain (Cogan, D. P. *et al. Science* 2021, 374, 729–734), likewise, our previous work (Zhai, G. *et al. Angew. Chem.*

Int. Ed. 2020, 56, 22738-22742) also suggested that KR_{1/2} plays an important role in stabilizing the conformation of ER_{1/2}. Accordingly, it is possible as you mentioned, that KR_{1/2} may facilitate the interaction between ER_{1/2} and module 3 in the native system. The lack of downstream modules would lead to the non-specific reaction on the chain extension product due to the increased dwell time of intermediates on the synthetase in the in vitro reaction system, for example, the uncommon iterative chain elongation of AurA occurred in the in vitro assay which was not found in vivo system (Busch, B. et al. *Angew. Chem. Int. Ed.* 2013, 125, 5393-5397). Similarly, the replacement of ER_{1/2} with other ER domains would result in a critical translocation delay of the unreduced product tethered onto ACP₃ under the strict substrate preference of the downstream gatekeeper. This would allow other ER domains to catalyze the non-specific activity in these ER domain replacement mutants. Together with the ER-KR didomain replacement experiment, we therefore thought that the downstream gatekeeper mechanism could be the more likely reason for accumulating a small amount of cross-module enoylreduction product rather than the direct interaction between KR_{1/2} and module 3.

We really appreciate your suggestion of using in vitro assay to test this hypothesis in the first round of review. However, the extremely low efficiency of cross-module enoylreduction of other ER domains, which were found in the in vivo system and the above factor 'non-specific reaction' greatly prevented us from performing the in vitro assay. Since there was no structural information to rule out the possibility and it does not influence the main conclusions of this manuscript, we have revised 'which ruled out the possible interaction between KR_{1/2} and module 3 in the cross-module enoylreduction' to 'which suggested other factors would be the reason for the accumulation of a small amount of cross-module enoylreduction product' in the revised manuscript to describe our assumption and avoid misleading.

REVIEWERS' COMMENTS

Reviewer #3 (Remarks to the Author):

All my concerns have been addressed.